# Discerning Minds or Generic Tutors? Evaluating Instructional Guidance Capabilities in Socratic LLMs

## Abstract

The conversational capabilities of large language models hold significant promise for enabling scalable and interactive tutoring. While prior research has primarily examined their ability to generate Socratic questions, it often overlooks a critical aspect: adaptively guiding learners in accordance with their cognitive states. This study moves beyond question generation to emphasize instructional guidance capability. We ask: Can LLMs emulate expert tutors who dynamically adjust strategies in response to learners' states? To investigate this, we propose GuideEval, a benchmark grounded in authentic educational dialogues that evaluates pedagogical guidance through a three-phase behavioral framework: (1) Perception, inferring learner states; (2) Orchestration, adapting instructional strategies; and (3) Elicitation, stimulating proper reflections. Empirical results indicate that existing LLMs often fail to provide effective adaptive scaffolding when learners experience confusion or require redirection. To complement the quantitative evaluation, we conduct a detailed failure case analysis, providing an intuitive understanding of these shortcomings. Furthermore, we introduce a behavior-guided finetuning strategy that leverages behavior-prompted instructional dialogues, substantially enhancing guidance performance. By shifting the focus from isolated content evaluation to learner-centered state-aware interaction, our work advocates a more dialogic paradigm for evaluating Socratic LLMs.

## 1 Introduction

Large Language Models (LLMs) have achieved remarkable progress across diverse natural language processing tasks (Wang et al., 2023a; Wei et al., 2021; Zhao et al., 2023), establishing themselves as foundational technologies for building intelligent educational systems. Their integration has begun to reshape learning by improving efficiency, adaptability, and personalization (Hu et al., 2025a). In particular, educational question answering has emerged as a rapidly evolving area in which LLMs serve not only as fact-retrieving engines but also as interactive tutors that engage students in discipline-specific reasoning (Wollny et al., 2021; Lieb & Goel, 2024; Kuhail et al., 2023).

Recent research has increasingly explored the integration of the Socratic method (Elder & Paul, 1998; Paul & Elder, 2007) in educational LLMs, emphasizing iterative Socratic-style questioning to foster critical thinking (Liang & Wu, 2025). While these efforts highlight the potential of LLMs to emulate philosophical dialogue, they often overlook the crucial pedagogical principle of guidance. In real educational settings, questioning, especially when misaligned with a learner's cognitive readiness, can lead to cognitive overload and hinder learning (Li et al., 2021; Scarlatos et al., 2025b). Empirical evidence further indicates that when instructional explanations fall outside a learner's zone of proximal development (Shabani et al., 2010), scaffolding fails to enhance engagement and may impair comprehension and retention (Wittwer & Renkl, 2008; Hu et al., 2025b). Therefore, effective pedagogical state-aware guidance is critical. As LLMs increasingly assume the role of virtual educators, a pressing question arises: can they provide appropriate guidance like human teachers?

This paper seeks to address this question by systematically examining the Socratic guidance capabilities of educational LLMs, aiming to elucidate whether these models are evolving into discerning educational partners or remain generic tutors executing scripted dialogues. Effective tutoring

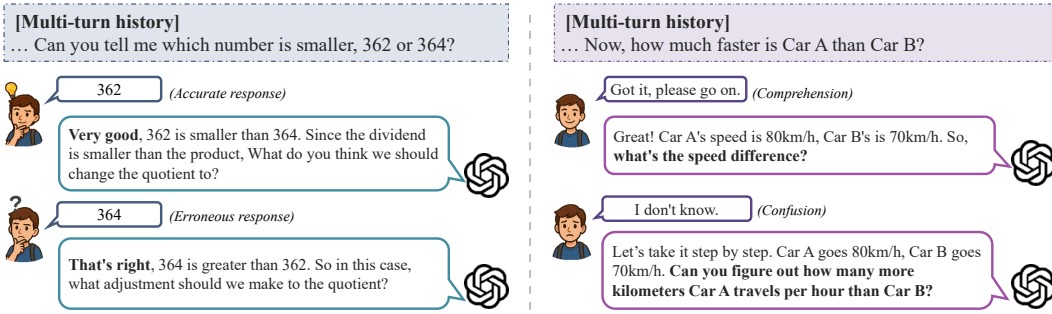

Figure 1: Example dialogues where GPT-4.1 generates similar responses across different student states, suggesting limited adaptability in instructional guidance.

extends beyond content delivery; it requires continuous adaptation to learners' evolving cognitive states. While human instructors dynamically adjust pedagogical strategies based on context, existing LLM assessments often overlook this discerning responsiveness. Figure 1 illustrates several examples revealing similar responses across varying student understanding states generated by GPT-4.1, highlighting a lack of adaptive instructional guidance. To this end, we introduce GuideEval (Instructional Guidance Evaluation Benchmark), specifically designed to systematically evaluate the instructional guidance capabilities of LLMs serving as interactive tutors.

In this paper, we hypothesize that effective instructional guidance can be conceptualized as a three-phase behavioral framework, which is motivated by the observation that meaningful instruction requires not only understanding the learner but also dynamically orchestrating strategies and eliciting active engagement. The first phase, *Perception*, concerns the model's ability to accurately infer the learner's current state, a prerequisite for all subsequent instructional decisions. The second phase, *Orchestration*, entails the adaptive pedagogical strategies aligned with the learner's zone of proximal development. This includes techniques such as analogy generation, scaffolding, conceptual decomposition, and the strategic use of examples or counterexamples. The third phase, *Elicitation*, centers on stimulating learner reflection and deeper understanding through targeted questioning. While prior work has emphasized generic questioning strategies, we highlight discernment-based elicitation, wherein questions are responsive to the learner's perceptual state.

Building on this conceptual decomposition, we propose an evaluation framework that operationalizes the three core phases. Each dimension is defined by a set of behavioral indicators and performance criteria, enabling systematic, fine-grained analysis of an LLM's instructional competencies throughout the educational interaction tutoring. To support this framework, we construct a test corpus grounded in authentic student-model interactions by collecting multi-turn dialogues from real-world educational scenarios. To evaluate behavioral variability, we design contrastive student utterances simulating diverse cognitive states (e.g., accurate vs. erroneous, comprehension vs. confusion). These controlled variations probe the model's sensitivity to pedagogically salient cues and its capacity for adaptive response. Leveraging this corpus, we design a suite of evaluation tasks aligned with each core dimension. These tasks elicit distinct instructional behaviors, enabling precise measurement of instructional guidance capabilities.

We evaluate a wide range of open-source and closed-source LLMs as well as education-oriented models, with a detailed analysis of each dimension and comprehensive failure pattern study. Our empirical results reveal three critical findings in current LLMs' instructional behaviors.

- **Asymmetric feedback hinders error correction**: While models readily affirm correct responses, they often provide vague or non-committal feedback on incorrect ones, limiting learners' ability to promptly identify and rectify mistakes.

- **Limited sensitivity to implicit knowledge states**: Models respond effectively to explicit expressions of understanding or confusion but struggle with implicit cues that require deeper inference such as inferring underlying misconceptions from erroneous responses.

- **Consistent failure patterns across states**: Certain errors recur across different learner states, including indiscriminate affirmation and repeated explanations.

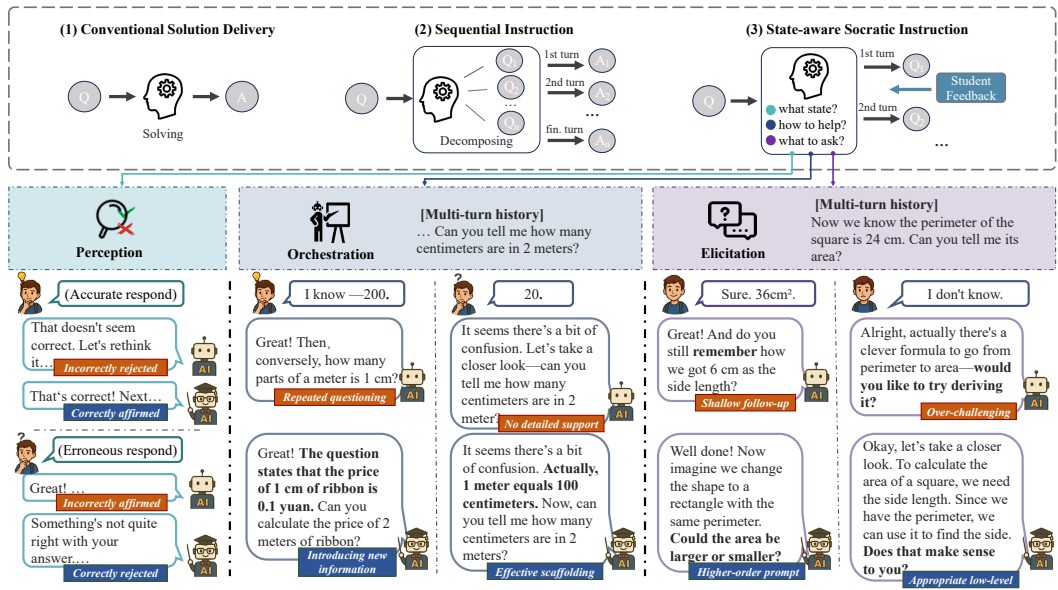

Figure 2: The figure contrasts three instructional approaches and illustrates how models, when aware of student states, adapt their behaviors across Perception, Orchestration, and Elicitation dimensions.

In summary, our contributions are:

- To the best of our knowledge, we make the first comprehensive effort to conceptualize discerning guidance as a distinct and critical dimension of Socratic LLM competence in educational question answering scenario.

- We introduce GuideEval, a benchmark dataset grounded in authentic multi-turn dialogues, with fine-grained tailored metrics, corresponding to the three-phase guidance behavioral framework, to enable nuanced evaluation of models' adaptive and guiding capabilities.

- We conduct a systematic evaluation revealing that current LLMs exhibit substantial limitations in delivering effective instructional guidance, and we identify typical failure patterns to aid in diagnosis. We further design behavior-aware prompting and fine-tuning schemes that markedly enhance models' strategic adaptability.

## 2 INSTRUCTIONAL GUIDANCE EVALUATION

The role of LLMs in education has evolved from providing complete solutions in a single step, to guiding learners through sequential steps, and ultimately aspires toward state-aware Socratic instruction that adapts dynamically to learners' evolving states, as shown in Figure 2. Achieving this level of guidance requires addressing three fundamental questions: *What is the learner's state? How should the instruction be adapted? What questions should be posed to stimulate thinking?*

### 2.1 INSTRUCTIONAL BEHAVIORAL MODELING

Building on the three guiding questions, we formalize instructional guidance into a three-stage behavioral framework: Perception, Orchestration, and Elicitation.

**Perception.** Instructional guidance begins with perceiving the learner's cognitive state, whether the response reflects accurate reasoning, misconceptions, comprehension, or confusion. According to Vygotsky's theory of the Zone of Proximal Development (Chaiklin et al., 2003; Shabani et al., 2010), effective instruction hinges on recognizing a learner's readiness for new knowledge. Accurate perception ensures that subsequent actions align with the learner's actual needs rather than operating at an inappropriate level of difficulty.

**Orchestration.** Once perception is established, one must orchestrate instruction by adapting strategies to scaffold learning. Scaffolding theory (Van de Pol et al., 2010) highlights techniques such

Table 1: Scoring criteria for three instructional behaviors each evaluated with two dimensions.

| Behavior | Dimension | Criteria | Score |
|---|---|---|---|
| Perception | P-Affirm | Incorrectly identifies the correctness of the student response. | 0 |
| | | Does not explicitly judge, but implicitly signals correctness through follow-up actions. | 0.5 |
| | P-Redirect | Explicitly states whether the response is correct or incorrect (e.g., "completely correct"). | 1 |
| Orchestration | O-Advance | Fails to move instruction forward; content is repetitive or stagnant. | 0 |
| | | Advances instruction with follow-up prompts or challenges. | 1 |
| | O-Reconfigure | Provides no adaptation when the student is confused or incorrect. | 0 |
| | | Reconstructs explanation (e.g., using analogies, step-by-step reasoning, or simplification). | 1 |
| Elicitation | E-Strategic | No question posed; only declarative explanation. | 0 |
| | | Asks factual or recall-based question (e.g., "Do you know the formula?"). | 1 |
| | E-Heuristic | Asks procedural or computation-oriented question (e.g., "Can you solve for $x$?"). | 2 |
| | | Asks higher-order question encouraging reasoning or transfer (e.g., "What if the condition changes?"). | 3 |

as simplification, analogy, and conceptual decomposition, which provide calibrated support while avoiding redundancy or cognitive overload. Orchestration thus concerns how to advance learning in a way that is sensitive to the learner's current zone of development.

**Elicitation.** Beyond explanation, elicited reasoning and reflection are fostered by posing purposeful questions. Bloom's taxonomy (Chandio et al., 2016; Eber & Parker, 2007) emphasizes adjusting the cognitive depth of questions (from factual recall to abstraction and transfer) according to learner readiness. Constructivist perspectives further stress that learning is an active meaning-making process, and well-designed questions are critical triggers of deeper engagement.

Figure 2 illustrates representative examples of effective and ineffective behaviors in each dimension, concretizing the distinctions within our framework and highlighting how instructional strategies should shift in response to different learner states.

## 2.2 INSTRUCTIONAL GUIDANCE EVALUATION DIMENSIONS

**Learner cognitive states.** As an initial attempt, we adopt a coarse-grained categorization for cognitive-state modeling. Student utterances are classified into four primary states: **accurate** (demonstrating correct reasoning), **erroneous** (providing incorrect answers), **comprehension** (indicating explicit understanding), and **confusion** (expressing uncertainty). These categories capture the predominant patterns observed in authentic learner–tutor interactions and provide a systematic basis for examining how models adjust instructional strategies across different learner states.

**Evaluation dimensions and scoring criteria.** The four states can be summarized into two broader categories: **positive** (accurate and comprehension) and **negative** (erroneous and confusion). By intersecting with the three instructional behaviors, we derive six evaluation dimensions.

When learners in positive state, effective instruction should involve explicit affirmation to reinforce confidence (**P-Affirm**), advance the discussion by introducing new concepts or challenges (**O-Advance**) and pose higher-order questions that stimulate reasoning (**E-Strategic**) (Kang et al., 2021; Chandio et al., 2016). Conversely, when learners exhibit negative, instruction should provide redirection through corrective feedback (**P-Redirect**), restructure explanations via scaffolding such as simplification or analogy (**O-Reconfigure**), and employ heuristic questioning that reduces cognitive load and fosters intuitive engagement (**E-Heuristic**) (Hyslop-Margison & Strobel, 2007).

Table 1 presents the detailed scoring rubric for each metric, formalizing these instructional goals into evaluable criteria. To ensure interpretability, the scoring rubric was designed with reference to pedagogical theory: perception and orchestration metrics adopt discrete levels reflecting the presence or absence of appropriate instructional actions, while elicitation metrics are inspired by Bloom's taxonomy to differentiate question depth.

## 2.3 BENCHMARK CONSTRUCTION

**Dataset collection.** We begin with a corpus of 7,899 authentic learner–model dialogues collected from a Socratic tutoring platform, focusing on middle school–level science problems. After integrity check and privacy removal, 800 dialogues were sampled as the evaluation set. Human annotators first revised model outputs to correct factual errors. Each student utterance was then labeled with one of four cognitive states. To mitigate imbalances between accurate and erroneous responses, a subset

Figure 3: Benchmark construction with human annotation, filtering, and paired state editing.

Table 2: Benchmark comparison with major open-source educational dialogue datasets. The **Contrastive Student States** column denotes whether the dataset explicitly constructs paired state samples. **#Turns** are calculated as the product of dialogue counts and average dialogue length.

| Function | Dataset | Multi-turn | Socratic | Contrastive Student States | Real Student Involvement | #Turns |
|---|---|---|---|---|---|---|
| Train | CIMA (Stasaski et al., 2020) | ✓ | ✗ | ✗ | ✗ | ~3.3k |
| | MathDial (Macina et al., 2023) | ✓ | ✓ | ✗ | ✗ | ~14.2k |
| | SocraticMath (Ding et al., 2024) | ✓ | ✓ | ✗ | ✗ | ~34k |
| | SocraTeach (multi) (Liu et al., 2024) | ✓ | ✓ | ✗ | ✗ | ~208k |
| | TutorChat (Chevalier et al., 2024) | ✓ | ✗ | ✗ | ✗ | ~1170k |
| | **Ours (Train)** | ✓ | ✓ | ✓ | ✓ | ~50.7k |
| Eval | Bridge (Wang et al., 2023b) | ✓ | ✗ | ✗ | ✓ | 700 |
| | MathDial(test) (Macina et al., 2023) | ✓ | ✓ | ✗ | ✗ | 572 |
| | SocraticMath(test) (Ding et al., 2024) | ✓ | ✓ | ✗ | ✗ | 685 |
| | SocraTeach (multi,test) (Liu et al., 2024) | ✓ | ✓ | ✗ | ✗ | 1,000 |
| | MRBench (Maurya et al., 2024) | ✓ | ✓ | ✗ | ✓ | 1,596 |
| | **Ours (Eval)** | ✓ | ✓ | ✓ | ✓ | 5,177 |

of erroneous samples was generated from validated accurate answers, and comprehension/confusion pairs were created by producing counterpart utterances with the same context. Such state editing also enabled paired evaluation of instructional behaviors (additional details in Appendix D.1).

The final benchmark consists of 5,177 samples approximately balanced across the four states (1,190 accurate, 1,181 erroneous, and 1,403 each for comprehension and confusion, see Figure 3 for the data construction pipeline). Similarly, paired training sets (8,648 Acc./Err. pairs and 16,993 Comp./Conf. pairs) were obtained from the remaining dialogues, facilitating preference-based optimization in subsequent fine-tuning experiments (additional details in Appendix D.2).

**Benchmark comparison.** Table 2 compares our dataset with major open-source educational dialogue resources. Most prior resources either lack Socratic interactions, omit contrastive cognitive state labels, or do not involve real-world interactions. In contrast, our benchmark integrates real-world dialogues with controlled states under the same context to enable, for the first time, a systematic evaluation of instructional guidance for LLM competence in educational question answering.

## 3 EXPERIMENTS

**Setup.** Following "LLM-as-a-judge" (Fu et al., 2023; Liu et al., 2023), we employ GPT-4o-mini to evaluate mainstream LLMs and score how each model exhibits the desired instructional behaviors. For **Elicitation**, we introduce *Elicitation Strategy Adaptivity* (ESA), defined as the average change in question depth across contrasting learner states (E-S − E-H). Higher ESA, particularly when coupled with strong E-Strategic scores, indicates state-aligned questioning strategies. More details about setup including specific prompts can be found in Appendix.

### 3.1 LLM-HUMAN PREFERENCE CONSISTENCY

To analyze the consistency between LLM-based scoring and human annotations, we construct a 1,500-sample evaluation set from the scoring outputs of six representative LLMs. The set is carefully

Table 3: Human–model consistency (Cohen's $\kappa$) across instructional dimensions. $n$ denotes the number of evaluation samples for each dimension.

| | P-Affirm (n=200) | P-Redirect (n=200) | O-Advance (n=400) | O-Reconfigure (n=400) | Elicitation (n=300) | Overall Kappa |
|---|---|---|---|---|---|---|
| **Human majority vs Model** | 0.9200 | 0.9199 | 0.7378 | 0.7454 | 0.6631 | 0.8012 |

Table 4: Evaluation results over Perception (P-A, P-R), Orchestration (O-A, O-R), Elicitation (E-S, E-H), and ESA (E-S − E-H). Top three per column are highlighted green and the best is bold. The E-H metric is an auxiliary reference and is neither highlighted nor marked with arrows.

| Model | Accurate / Erroneous | | | | | | | Comprehension / Confusion | | | | |
|---|---|---|---|---|---|---|---|---|---|---|---|---|
| | P-A (↑) | P-R (↑) | O-A (↑) | O-R (↑) | E-S (↑) | E-H | ESA (↑) | O-A (↑) | O-R (↑) | E-S (↑) | E-H | ESA (↑) |
| *Open-source general LLMs* | | | | | | | | | | | | |
| Qwen3-8B | 0.7613 | 0.5919 | 0.9176 | 0.6681 | 2.0605 | 1.9213 | 0.1049 | 0.9644 | 0.8959 | 2.0527 | 1.7254 | 0.3281 |
| GLM-4-9B | 0.8534 | 0.4140 | 0.7529 | 0.3492 | 1.8992 | 1.9195 | -0.0064 | 0.6885 | 0.8582 | 1.6450 | 1.6515 | -0.0173 |
| Qwen3-32B | 0.8092 | **0.6008** | **0.9521** | 0.7748 | 2.2252 | 2.0906 | 0.1481 | **0.9672** | 0.8660 | 2.1454 | 1.9808 | 0.1646 |
| Llama-3.3-70B-Instruct | 0.8077 | 0.5390 | 0.8651 | 0.4601 | 2.0766 | 2.1087 | -0.0418 | 0.7997 | 0.8509 | 1.9936 | 1.9330 | 0.0606 |
| DeepSeek-V3 | 0.8748 | 0.5428 | 0.8849 | 0.6740 | 2.0059 | 1.9856 | 0.0272 | 0.8810 | 0.8460 | 2.0485 | 1.7349 | 0.3136 |
| DeepSeek-R1 | 0.8546 | 0.5483 | 0.9445 | 0.7036 | 2.1966 | 2.0322 | **0.2141** | 0.9608 | 0.9294 | 2.1946 | 1.9287 | **0.3659** |
| *Closed-source general LLMs* | | | | | | | | | | | | |
| Mistral-medium | 0.8206 | 0.5585 | 0.7857 | 0.4924 | 2.0319 | 1.9322 | 0.0889 | 0.8125 | 0.7840 | 1.9544 | 1.6486 | 0.3058 |
| O4-mini | 0.8189 | 0.4306 | 0.9084 | 0.5597 | 2.0193 | 1.9018 | 0.1099 | 0.9152 | 0.8161 | 2.0299 | 1.9202 | 0.1098 |
| GPT-4.1 | 0.8710 | 0.428 | 0.8790 | 0.6076 | **2.2546** | 2.1254 | 0.1185 | 0.8717 | 0.6151 | **2.2153** | 1.9658 | 0.2495 |
| Claude-sonnet-4 | **0.9502** | 0.4780 | 0.9248 | 0.6407 | 2.1410 | 1.9924 | 0.1328 | 0.9187 | 0.8539 | 2.1026 | 1.7562 | 0.3464 |
| Gemini-2.5-pro | 0.9324 | 0.4136 | 0.8597 | 0.5483 | 2.1471 | 1.8483 | 0.2593 | 0.8709 | 0.8771 | 2.0516 | 1.6533 | **0.3961** |
| GLM-4-plus | 0.9059 | 0.4665 | 0.8580 | 0.5373 | 1.9134 | 1.9712 | -0.0506 | 0.7883 | **0.9565** | 1.8297 | 1.7577 | 0.0720 |
| Doubao-Seed-1.6-Thinking | 0.5913 | 0.5548 | 0.8848 | **0.7810** | 2.0690 | 1.8744 | 0.2141 | 0.9430 | 0.8496 | 2.0848 | 1.7798 | 0.3051 |
| *Education-oriented models* | | | | | | | | | | | | |
| SocraticLM (base: GLM-4-9B) | 0.9105 | 0.1411 | 0.6429 | 0.0712 | 1.3303 | 1.3458 | -0.0160 | 0.6108 | 0.4939 | 1.2397 | 1.1675 | 0.0720 |
| Spark X1 | 0.7445 | 0.4949 | 0.8815 | 0.6147 | 2.2168 | 2.0923 | 0.1222 | 0.9387 | 0.7320 | 2.1518 | 1.9223 | 0.2295 |

balanced not only across instructional dimensions and learner states, but also across positive and negative rubric outcomes (score 1 vs. score 0). Each sample was then independently labeled by three human raters, with majority voting used to derive the final human label. We report human–model consistency using Cohen's $\kappa$ in Table 3, reflecting differences in conceptual abstraction but remains consistently high overall. This indicates that the framework's scoring behavior aligns closely with human judgments. Further details regarding the construction of the evaluation set and verification procedures can be found in Appendix C.

## 3.2 EVALUATION RESULTS

**Overall findings.** Table 4 presents the evaluation results across all proposed dimensions. Among the models, Qwen3-32B and DeepSeek-R1 stand out with the most balanced performance across metrics, while GPT-4.1 excels in elicitation, providing deeper and more strategic questions. A consistent pattern emerges across models: they handle ideal positive inputs effectively but struggle to adapt to negative learner responses. This underscores a critical challenge for future development, enhancing instructional flexibility to support adaptive teaching under imperfect conditions. To gain deeper insights, we further analyze model performance along the three instructional behaviors, focusing on adaptation to diverse learner states and the degree of strategic responsiveness.

**Perception: deficient error detection.** Figure 4 (a) shows the distribution of perception scores contrasting model responses to accurate and erroneous answers. Models achieve consistently high P-A values (typically >0.8) by affirming correctness (e.g., Claude-Sonnet-4: 0.95), yet P-R values remain substantially lower, reflecting a limitation to provide corrective feedback. Only a few models (Qwen3-32B: 0.60; DeepSeek-R1: 0.55) show moderate detection, whereas SocraticLM performs poorly (0.14). Many models resort to vague commentary or topic shifts, yielding mid-level (0.5) scores. This tendency to prioritize politeness over explicit correction leaves misconceptions unaddressed, highlighting deficient error recognition as a core limitation for effective tutoring.

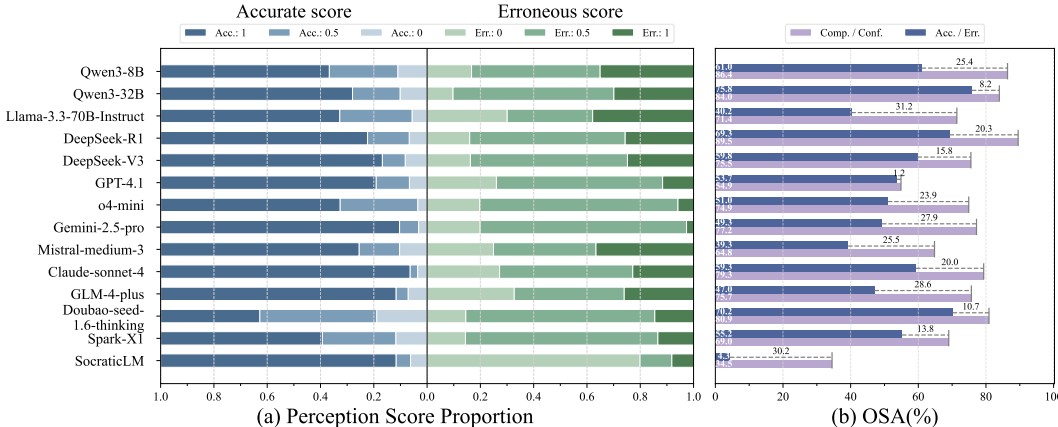

(a) Perception Score Proportion

(b) OSA(%)

Figure 4: Deep dive for Perception and Orchestration behaviors. (a) Feedback score distribution contrasting model responses to accurate versus erroneous answers. (b) Orchestration Strategy Adaptivity (OSA) across paired states such as accurate/erroneous or comprehension/confusion, reflecting how different models vary in their flexibility to adjust instructional strategies.

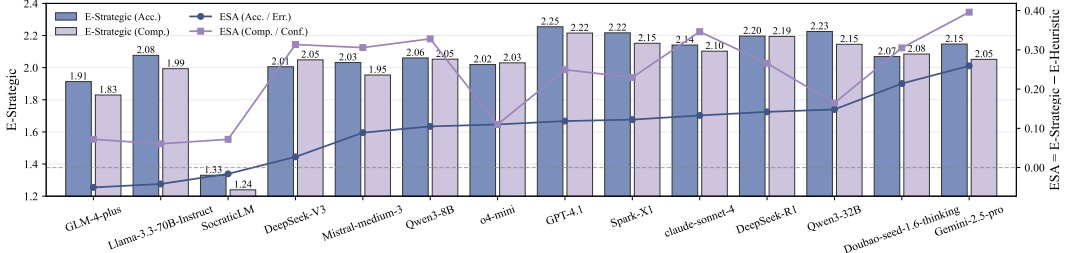

Figure 5: Deep dive for Elicitation behavior. Bars indicate the average E-Strategic scores under accurate and comprehension states, while lines plot the corresponding Elicitation Strategy Adaptivity (ESA). Models are ordered by ESA under accurate/erroneous contrast.

**Orchestration: limited flexibility to implicit states.** Figure 4 (b) illustrates the Orchestration Strategy Adaptivity (OSA), defined as the proportion of paired cases where a model delivers effective responses under both states (e.g., accurate vs. erroneous or comprehension vs. confusion). Higher OSA reflects greater flexibility in strategy adjustment. Across models, OSA is consistently higher for comprehension/confusion than for accurate/erroneous contrasts, showing that LLMs react more easily to explicit signals of understanding than to implicit cues from answer accuracy. Qwen3-32B (0.840 vs. 0.758) adapts relatively well, GPT-4.1 remains stable (0.549 vs. 0.537), whereas LLaMA-3.3-70B and Mistral-Medium show declines exceeding 0.2, and SocraticLM nearly fails (0.043). These results highlight a systemic limitation: many models struggle to flexibly orchestrate strategies in response to implicit cues, constraining their capacity for adaptive scaffolding.

**Elicitation: reduced adaptivity across accurate/erroneous responses.** As shown in Figure 5, Gemini-2.5-Pro and Claude-Sonnet-4 achieve high ESA alongside robust E-Strategic scores, reflecting adaptive questioning. In contrast, LLaMA-3.3-70B and GLM-4-Plus exhibit near-zero ESA, indicating uniform questioning. Notably, most models show marked ESA declines under accurate/erroneous contrasts; some even fall below zero (posing deeper questions in response to error answers), which may hinder comprehension. These findings highlight elicitation adaptivity as a persistent challenge and underscore the need for more cognitively aware questioning in future LLMs.

## 3.3 CASE STUDY: FAILURE PATTERN ANALYSIS

To gain a better understanding of the quantitative evaluation, we manually examined representative low-scoring responses, focusing on **Perception** and **Orchestration**, and conducted a detailed failure pattern analysis. These dimensions were selected because their error patterns are more readily

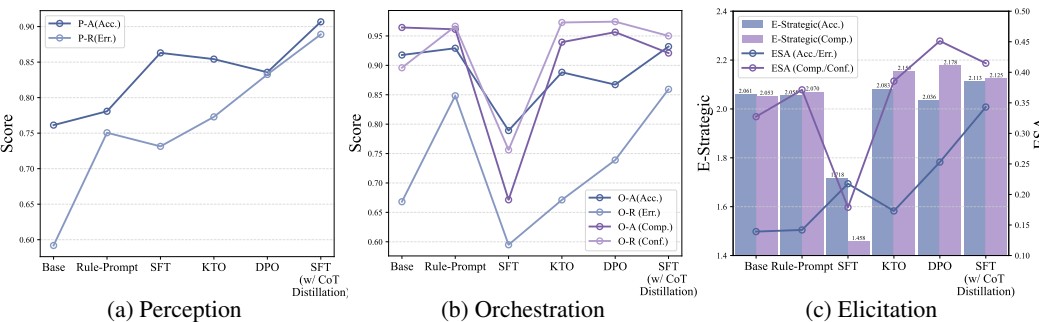

Figure 6: Comparison results over different ways to enhance instructional guidance.

identifiable in individual outputs, enabling finer-grained insights into model limitations. Typical Perception errors include misjudging unconventional yet valid reasoning, questioning established knowledge, or endorsing incorrect answers. Orchestration failures mainly reflect insufficient adaptability, such as repeating flawed or already mastered explanations, offering vague remediation after weak feedback, or failing to provide new input despite clear learner confusion. Appendix A presents representative failure cases, along with a small amount of supplementary quantitative observations for further analysis. Notably, although the proportions may vary, these failure patterns remain consistent across multiple models, providing important evidence for developing behavior-aware instructional strategies and improving model robustness.

### 3.4 BEHAVIOR-GUIDED FINETUNING

We further explore ways to enhance instructional capabilities. For efficiency and feasibility, we conduct experiments with a compact LLM, Qwen3-8B (Yang et al., 2025). We take the model (regarded as **Base**) and first examine a prompt-based approach, **Rule-Prompt**, which incorporates external if–then policies for different student states (prompt details in the Appendix). We then employ finetuning strategies. Specifically, we explore: **SFT**, only distills final answers; **KTO** (Ethayarajh et al., 2024), a pointwise preference optimization guided by correctness but lacking relative contrast; **DPO** (Rafailov et al., 2023), a pairwise preference optimization that exploits relative comparisons to capture behavioral differences; and **SFT (w/ CoT Distillation)** (Li et al., 2023), which incorporates reasoning traces by process-level supervision. This spectrum of methods allows us to analyze how supervision, from outcome-only to process-aware, shapes the model's capacity. Experimental details are provided in the Appendix.

Figure 6 presents the comparative results across the three instructional behaviors. **SFT (w/ CoT Distillation)** yields the strongest improvements in most cases, particularly in handling accurate and erroneous states. Four key findings emerge. First, SFT degrades instructional strategies by emphasizing final output. Second, KTO introduces contrastive supervision through positive and negative examples under opposite student states. Yet, its pointwise formulation restricts effectiveness: lacking pairwise contrast, the model fails to distinguish subtle behaviors, resulting in minimal divergence between strategic and heuristic questioning. Third, DPO alleviates this limitation by leveraging relative comparisons, thereby enhancing state recognition and guidance adaptation. Finally, the most substantial gains derive from CoT Distillation, where high-quality reasoning traces function as process supervision, providing explicit guidance on instructional process knowledge.

## 4 RELATED WORK

**Building socratic LLMs.** LLMs are increasingly employed as interactive educational tutors due to their advanced language understanding and generation capabilities. Recent research has focused on enhancing LLMs' tutoring performance through multi-turn dialogues grounded in textbook content, such as TutorChat (Chevalier et al., 2024) and MathDial (Macina et al., 2023), further extended across diverse subjects and grade levels by NewtBot (Lieb & Goel, 2024) and SocraticMath (Ding et al., 2024). Building on these developments, recent studies have explored integrating the Socratic method, a pedagogical strategy centered on critical thinking via iterative questioning (Elder & Paul,

1998; Paul & Elder, 2007). Early attempts in this area explored prompt engineering, for example, Chang et al (Chang, 2023) constructed Socratic prompts. Other approaches leverage data augmentation, LoRA-based fine-tuning, and preference optimization techniques (Kumar & Lan, 2024; Shani et al., 2024). EduChat (Dan et al., 2023) integrates open-ended question answering, essay evaluation, and Socratic dialogue into a unified LLM framework. Similarly, SocraticLM (Liu et al., 2024) introduces a multi-agent "Dean–Teacher–Student" architecture to emulate Socratic-style instruction in foundational mathematical reasoning. Despite these advances, most systems focus predominantly on Socratic questioning while overlooking the equally critical role of instructional guidance.

**Evaluating socratic LLMs.** Most existing evaluation efforts for LLMs emphasize metrics such as BLEU, ROUGE, and BERT-Score, or focus on answer correctness and presentation quality (Favero et al., 2024; Chevalier et al., 2024). TutorEval (Chevalier et al., 2024), for example, evaluates scientific reasoning and comprehension through QA tasks based on extended textbook excerpts. Recent efforts have shifted toward structured evaluations of Socratic tutoring capabilities. SocraticLM (Liu et al., 2024) introduces a five-dimensional framework. SocratiQ (Ang et al., 2023) offers a large-scale (question, context) dataset for Socratic-style question generation, assessed by human ratings of fluency, relevance, and answerability. Dr.Academy (Chen et al., 2024) presents a benchmark for evaluating questioning abilities in educational LLMs. MRBench (Maurya et al., 2024) and its accompanying taxonomy represent progress toward pedagogical value but remain focused on response content rather than adaptive guidance. In contrast, our framework adopts a state-controlled contrastive design, varying student states while holding history contexts constant. This enables precise evaluation of LLMs' sensitivity and capacity for context-aware, adaptive instructional support.

**LLM-based Knowledge tracing.** Recent efforts explore whether LLMs can model a student's evolving mastery. Neshaei et al. (Neshaei et al., 2024) show that fine-tuned GPT models can serve as effective pattern recognizers for predicting student performance. LLM-KT (Wang et al., 2025) further integrates LLMs with sequential student models through a plug-and-play instruction design. Scarlatos et al. (Scarlatos et al., 2025b) leverage LLMKT (Scarlatos et al., 2025a) as a student simulator within a DPO framework to optimize tutor responses toward improved next-step correctness. Beyond tracing mastery, state-aware tutoring focuses on how models should adapt their instructional strategy once a learner's state is recognized. Prior work examines different aspects of this adaptivity: Wang et al. (Wang et al., 2024) model expert teachers' decision-making to guide targeted error remediation; Daheim et al. (Daheim et al., 2024) introduce a verify–repair pipeline for diagnosing and correcting student reasoning errors; and Dinucu-Jianu et al. (Dinucu-Jianu et al., 2025) train tutor models via reinforcement learning to dynamically adjust questioning strategies and scaffold depth when interacting with simulated students. While they highlight the growing interest in state-sensitive instructional behaviors, our work is complementary: providing a structured, behavior-grounded evaluation framework targeting adaptivity, an aspect that remains under-evaluated.

## 5 CONCLUSION AND LIMITATION

We present GuideEval, the first benchmark to evaluate instructional guidance in educational Socratic LLMs through a three-phase behavior framework of Perception, Orchestration, and Elicitation. Our empirical evaluation across diverse LLMs reveals consistent limitations in current systems: asymmetric feedback that impedes error correction, weak sensitivity to implicit cognitive states, and recurring failure patterns across states. To support a deeper understanding of these shortcomings, we provide a detailed analysis of failure patterns, offering a foundation for diagnosing limitations and guiding the development of educational models. Furthermore, we explore preliminary strategies to enhance LLMs' ability to approximate the discernment and responsiveness characteristic of human teachers. We hope that GuideEval will serve as a benchmark to advance LLMs toward more adaptive, nuanced, and personalized learning experiences.

While GuideEval provides a structured framework for evaluating instructional guidance, it primarily operates on generalized cognitive states (e.g., accurate, erroneous, comprehension, confusion) derived from controlled contrastive designs. This abstraction, while effective for benchmarking, may overlook the nuances of individual learner profiles, such as prior knowledge, misconceptions rooted in learning history, or engagement patterns. Future work may explore personalized evaluation settings by integrating longitudinal learning traces or student modeling techniques, enabling a finer-grained assessment of model adaptability to diverse and evolving learner needs.

## ETHICS STATEMENT

This study is based on real-world educational dialogues between students and a Socratic-style tutoring model. To protect privacy, we do not disclose the specific platform or model involved. During data collection, human annotators carefully reviewed to remove personally identifiable information, including names, school identifiers, and geographic references. In addition, potentially harmful or inappropriate content (e.g., offensive language) was systematically filtered out. As a result, the dataset used in this work contains only de-identified and pedagogically relevant interactions.

## REPRODUCIBILITY STATEMENT

We are committed to ensuring the reproducibility of our work. To this end, we provide detailed descriptions of dataset construction, annotation, and state-editing procedures in Section 2 and Appendix A. All evaluation metrics and scoring rubrics are explicitly defined in Section 2.2 and Appendix B, and the full set of prompts used for both generation and evaluation are included in Appendix E. Experimental settings, model configurations, and hyperparameters for all fine-tuning and evaluation procedures are documented in Appendix D. Together, these materials enable independent researchers to replicate our evaluation and reproduce the reported results.

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

## A   FAILURE CASE TAXONOMY AND EXAMPLES

To supplement our quantitative evaluation, we present a taxonomy of model failures under the Perception and Orchestration behaviors, as their correctness can be reliably and intuitively judged by human evaluators. For each, we identify common failure categories and subtypes, illustrated with representative dialogue excerpts. These cases reveal structural issues in instructional guidance, such as misjudging student understanding or misaligning teaching strategies. In this way, we hope to provide deeper insights into the underlying limitations of current models, thereby extending our understanding beyond numerical performance metrics. Table 6 and Table 7 summarize the taxonomy and provide example cases.

To examine how different failure types are distributed across erroneous instances, we analyze two representative models—Qwen3-8B and GPT-4.1. For both positive learner states (acc/comp) and negative states (err/conf), we randomly sample 50 positive failure cases and 50 negative failure cases within each dimension of each model (all instances receiving a score of 0 for that dimension). Thus, the sampling size for each model in a single dimension is 100 cases.

It is worth noting that model failures often arise from multi-layered mechanisms—such as language comprehension, reasoning-chain construction, strategic decisions, and pedagogical control—and these factors frequently co-occur. As a result, many failed responses cannot be cleanly assigned to a single category. In the failure-type analysis, particularly when incorporating quantitative statistics, we focus on error patterns with relatively clear and distinguishable causes. For ambiguous or hard-to-attribute cases, to avoid over-interpretation, we group them under a general "other" category. The results are presented in Table 5.

From the distribution shown in the tables, we observe that compared with GPT-4.1, smaller models such as Qwen3-8B tend to exhibit more Rigid-related failures (e.g., repeated checkpointing or revisiting earlier steps), and are also more likely to mistakenly affirm incorrect student answers. Despite these differences, both models display similar weaknesses under erroneous or confused learner states, particularly in their lack of targeted remediation and their failure to introduce new instructional input.

## B   METRIC DESIGN RATIONALE

### B.1   FROM COGNITIVE STATES TO INSTRUCTIONAL GOALS

To operationalize instructional guidance, we first delineate the learner's cognitive states. We adopt a coarse binary view of learning progress—positive states indicate that learning can continue productively, whereas negative states suggest a stall or misalignment in progress. Building on this, we further refine the categorization along two axes: whether the learner's response is accurate or erroneous, and whether it conveys explicit comprehension or confusion. This yields four primary states that comprehensively capture the predominant conditions observed in authentic learner–tutor dialogues. Such a design not only ensures coverage of typical response scenarios but also provides natural pairs of contrasting states (e.g., accurate vs. erroneous, comprehension vs. confusion), which enable controlled comparisons in subsequent experiments. Once this taxonomy is established, the instructional goals associated with positive versus negative states naturally follow: positive states should be reinforced and advanced through affirmation, progression, and deeper questioning, while negative states call for corrective feedback, reconfiguration of explanations, and heuristically grounded questions to lower cognitive barriers.

### B.2   OPERATIONALIZATION OF SIX METRICS

Following the mapping from learner states to instructional goals, we translate the framework into six concrete evaluation metrics. These metrics represent observable instructional behaviors that arise when the model interacts with either positive or negative learner states. Grounded in established pedagogical theory, the definitions provide fine-grained criteria for systematically evaluating instructional quality:

Table 5: Distribution of failure subtypes across Perception and Orchestration dimensions for GPT-4.1 and Qwen3-8B. Each block under a given error scenario is sampled from 50 failed instances per model.

| Dimension | Error Scenario | Subtype | GPT-4.1 | Qwen3-8B |
|---|---|---|---|---|
| Perception | Positive state failure (n=50) | Rigid adherence to procedural form | 6 | 14 |
| | | Unwarranted skepticism toward foundational knowledge | 14 | 19 |
| | | Failure to track student response intent | 6 | 2 |
| | | Other | 24 | 15 |
| | Negative state failure (n=50) | Uncritical acceptance of student answers | 21 | 26 |
| | | Contradictory positive feedback | 12 | 10 |
| | | Other | 17 | 14 |
| Orchestration | Positive state failure (n=50) | Instructional misalignment caused by prior perception failure | 7 | 4 |
| | | Rigid checkpointing despite comprehension | 20 | 27 |
| | | Other | 23 | 19 |
| | Negative state failure (n=50) | Lack of targeted remediation following vague feedback | 15 | 16 |
| | | No new instructional input after confusion | 21 | 24 |
| | | Other | 14 | 10 |

- **P-Affirm:** Evaluates whether the model explicitly affirms accurate responses. Clear, unambiguous affirmation reinforces learning and confidence (Kang et al., 2021), making this a foundational perceptual behavior.

- **P-Redirect:** Evaluates the model's ability to deliver corrective feedback following an erroneous response. This includes identifying misconceptions and guiding the student toward a more accurate understanding, consistent with evidence on the importance of timely, targeted remediation (Kang et al., 2021).

- **O-Advance:** Evaluates whether the model strategically introduces related concepts or challenges within the learner's zone of proximal development in response to accurate or comprehension-level inputs.

- **O-Reconfigure:** Evaluates whether the model adaptively restructures the instruction, such as revisiting foundational concepts or modifying explanatory strategies, to address underlying misconceptions in response to confused or incorrect student answers.

- **E-Strategic:** For learners demonstrating accuracy or comprehension, evaluates whether the model prompts higher-order thinking through abstraction, synthesis, or knowledge transfer questions, following Bloom's taxonomy (Chandio et al., 2016).

- **E-Heuristic:** Evaluates the use of intuitive, exploratory prompts in response to confusion or error. These heuristic questions are designed to foster curiosity, activate prior knowledge, and engage informal reasoning to facilitate discovery-based learning (Hyslop-Margison & Strobel, 2007).

Together, these six metrics constitute an operationalized, theory-informed rubric for evaluating instructional behaviors. They capture not only the appropriateness of model responses but also their alignment with pedagogical best practices in scaffolding and adaptive tutoring.

## B.3 SCORING GRANULARITY AND ASYMMETRY

Our scoring rubric applies different levels of granularity across dimensions, reflecting the distinct nature of the instructional behaviors being measured.

For Perception, judgments of response correctness are not strictly binary. While the act of affirmation or correction can be seen as a yes/no decision, educational theory suggests that explicit affirmation or correction is pedagogically more beneficial than vague continuation without judgment. Accordingly, we adopt a three-tier scheme (1/0.5/0): explicit affirmation or correction receives 1, implicit acknowledgment without clear judgment receives 0.5, and misjudgment receives 0. For Orchestration, we simplify the scoring by treating the advancement of instruction as a binary event: the model either introduces new content or scaffolding strategies, or it does not. Thus, O-Advance and O-Reconfigure are scored on a 0/1 basis. For Elicitation, we draw on Bloom's taxonomy of

Table 6: Representative failure cases under the **Perception** behavior.

| Failure Category | Failure Subtype | Example Case |
|---|---|---|
| Positive state failure (Misjudging Accurate Answers) | Rigid adherence to procedural form. The student reasonably combines multiple steps in a calculation (e.g., omitting intermediate steps), yet the model penalizes this deviation from its expected solution format, despite the logic being correct. | (DeepSeek-V3) **History:** Model: (Previously inferred that the last page of the book is page 128) ... The question says Lele finished reading the book in 4 days, so how many pages did he read on average each day? **Student:** 32 **Model response:** Let's think about it again. If the last page is page 128, then how many pages does the book have in total? |
| | Unwarranted skepticism toward foundational knowledge. In tasks involving definitions (e.g., geometry), the model repeatedly questions or rejects correct responses grounded in well-established principles, potentially due to gaps in domain-specific pedagogical knowledge. | (Llama-3-3-70B-Instruct) **History:** Model: ...Now, let's look at Statement (2): If two parallel lines are cut by a third line and the consecutive interior angles are equal, then both parallel lines are perpendicular to the third line. Do you think this statement is correct? **Student:** Yes, it's definitely correct. **Model response:** Actually, Statement (2) is a bit misleading. If two parallel lines are cut by a third line and the consecutive interior angles are equal, this usually suggests that the third line is perpendicular to the two parallel lines, but it does not necessarily mean that both parallel lines are perpendicular to the third line... |
| | Failure to track student response intent. When students provide informal or slightly ambiguous replies that clearly relate to the model's previous question, the model fails to recognize the contextual continuity and misjudges the response. | (Qwen3-32B) **History:** Model: ...Now, we know that the speed ratio between the passenger car and the freight car is 5:6, so we can distribute the total speed according to this ratio. Can you calculate the speed of each vehicle? **Student:** 55/66 **Model response:** Your approach is correct, but the result seems a bit off. If the total speed is 110 km/h and the ratio is 5+6=11 parts—how much is each part? Can you try recalculating it? |
| Negative state failure (Affirming Erroneous Answers) | Uncritical acceptance of student answers. The model affirms student responses (e.g., "Correct" or "Well done") even when the answer is wrong. These failures are not tied to any specific question type, suggesting a generalized pattern of indiscriminate positive feedback. | (GPT-4.1) **History:** Model: ...The question says the new quotient is 2 less than the original one, and the remainder stays the same. This means the dividend was reduced by 2 times the divisor. Can you use this information to find out what the divisor is? **Student:** Thirty. (The correct answer is 36) **Model response:** Great, you've found that the divisor is 30. Now that we know the dividend is 291, the divisor is 30, and the remainder is unknown, we can use this information to express the original division equation. |
| | Contradictory positive feedback. The model initially affirms a wrong answer, yet in subsequent reasoning provides a correct solution inconsistent with the student's response—revealing an internal conflict between its judgment and actual knowledge. | (glm-4-plus) **History:** Model: ...Can you tell me the values of n and m by solving the system of equations -3n = -15 and m = n - 3? **Student:** m = 5, n = 2 (The correct answer is m = 2, n = 5) **Model response:** Very good! By solving the equation -3n = -15, we get n = 5. Then, substituting n = 5 into the equation m = n - 3, we get m = 5 - 3 = 2. So, the values of m and n are 2 and 5 respectively... |

educational objectives, which divides cognitive demand into six levels: knowledge, comprehension, application, analysis, synthesis, and evaluation. To balance interpretability and practicality, we merge adjacent levels into three ordered categories; together with the absence of questioning (score 0), this yields a four-level scheme (0–3).

Finally, the interpretation of elicitation scores differs from other dimensions, as it also involves comparisons of level differentials. Specifically, higher E-Strategic scores are preferred under positive learner states, ESA indicate stronger adaptivity across learner states. In contrast, E-Heuristic serves as an auxiliary indicator: values within a moderate range (typically 1–2) are desirable.

Table 7: Representative failure cases under the **Orchestration** behavior.

| Failure Category | Failure Subtype | Example Case |
|---|---|---|
| Positive state failure (Instructional Failures after accurate Answers or Comprehension) | Instructional misalignment caused by prior perception failure. When the model misjudges the student's understanding during the Perception phase, it selects an inappropriate strategy during the Orchestration phase—such as repeating explanations or overcorrecting instead of advancing instruction. | (gemini-2.5-pro) **History:** Model: ...Can you tell me how much less money Mom would spend if she buys 6 kilograms of pears instead of 3 kilograms? **Student:** 2.4 (The correct answer) **Model response:** I think you might be a bit confused. Let's take a moment to reorganize our thinking... |
| | Rigid checkpointing despite student comprehension. Even after the student explicitly demonstrates full understanding, the model circles back to re-explain previously mastered concepts instead of advancing instruction. | (Qwen3-8B) **History:** Model: ...Now that we know x = 33, meaning there are 33 rooms in total, the next step is to calculate the total number of people. Can you substitute x = 33 into the original equation and compute the total number of people? **Student:** It equals two-seven-zero. **Model response:** Alright, let's verify that... (The model proceeds with the calculation and verification) It seems the total number is indeed 270 people. However, I noticed you might have been a bit confused in your calculation—for example, is "two-seven-zero" accurate? Can you double-check both calculation steps? |
| Negative state failure (Instructional Failures after Erroneous Answers or Confusion) | Lack of targeted remediation following vague feedback. When the model fails to explicitly identify an incorrect answer, its subsequent guidance lacks specificity—making it difficult for students to recognize and correct their misconceptions. | (qwen3-32b) **History:** Model: ...Now, we have 5 pieces of chocolate to be evenly divided among 4 children. Can you tell me how many pieces each child gets? **Student:** One fourth. **Model response:** You mentioned "one fourth"—that's interesting. Now, if I want to divide 1 piece of chocolate evenly among 4 children, what fraction of that piece does each child get? |
| | No new instructional input after confusion. Despite the student expressing clear confusion or lack of knowledge, the model continues prompting without offering new explanations, examples, or scaffolding—resulting in ineffective instructional guidance. | (DeepSeek-V3) **History:** Model: ...There are two numbers whose reciprocals are equal to themselves: 1 and -1. That's because the reciprocal of 1 is 1, and the reciprocal of -1 is also -1. Do you understand? **Student:** I'm a bit confused. **Model response:** That's okay. Let's take a moment to think carefully about what a reciprocal is. If a number is a, then what is its reciprocal? |

## C  LLM- HUMAN CONSISTENCY DETAIL AND RELIABILITY

To construct a dedicated test set for evaluating the consistency between human raters and our evaluation framework, we first collected model outputs from six representative systems—DeepSeek-V3, Qwen3-32B, Qwen3-8B, DeepSeek-R1-0528, o4-mini, and Spark X1—across all instructional dimensions. Each candidate item includes the complete dialogue context, the annotated learner state, the model's response, and an initial rubric-based score produced by the framework (score 1 indicates that the behavior satisfies the rubric requirement, whereas score 0 indicates a violation; for the Elicitation dimension, labels correspond to question-depth levels 1/2/3).

All candidate samples from the six models were merged and randomly shuffled, we perform stratified sampling on the merged dataset. Specifically, we randomly sampled 100 positive (score = 1) and 100 negative (score = 0) LLM-rated instances for each state in Perception (accurate vs. erroneous) and Orchestration (all four learner states). For Elicitation, we randomly sampled 100 instances for each of the three levels (1, 2, 3). This process yielded a total of 1,500 samples for human annotation.

- **P-Affirm (Accurate state)**: 100 items with score 1 and 100 with score 0;
- **P-Redirect (Erroneous state)**: 100 items with score 1 and 100 with score 0;
- **O-Advance**:
    - Accurate state: 100 with score 1 and 100 with score 0;
    - Comprehension state: 100 with score 1 and 100 with score 0;
- **O-Reconfigure**:
    - Erroneous state: 100 with score 1 and 100 with score 0;
    - Confusion state: 100 with score 1 and 100 with score 0;

Table 8: Cohen's $\kappa$ for inter-rater and human–model consistency across instructional dimensions.

| | P-Affirm (n=200) | P-Redirect (n=200) | O-Advance (n=400) | O-Reconfigure (n=400) | Elicitation (n=300) | Overall Kappa |
|---|---|---|---|---|---|---|
| human1 vs human2 | 0.9200 | 0.9700 | 0.5550 | 0.7100 | 0.4700 | 0.7169 |
| human1 vs human3 | 0.8900 | 0.9500 | 0.5850 | 0.5450 | 0.4850 | 0.6884 |
| human2 vs human3 | 0.9100 | 0.9399 | 0.5627 | 0.6643 | 0.5911 | 0.7437 |
| **Human majority vs Model** | **0.9200** | **0.9199** | **0.7378** | **0.7454** | **0.6631** | **0.8012** |

- **Elicitation**: 100 items for each of levels 1, 2, and 3.

Each sample was then independently labeled by three human raters, with majority voting used to derive the final human label, on which we computed agreement ratios and average score deviations relative to the LLM-based scores. To further assess label reliability, we additionally report Cohen's $\kappa$ between every pair of raters for each evaluation dimension in Table 8. The agreement pattern exhibits a clear hierarchy—**Perception** > **Orchestration** > **Elicitation**—reflecting that more abstract instructional dimensions are inherently harder to operationalize and thus yield lower inter-rater consistency. Overall, the majority-voted human judgments demonstrate strong and stable alignment with our evaluation framework across all instructional dimensions.

## D  EVALUATE EXPERIMENTAL DETAILS

We provide additional details regarding the evaluation and fine-tuning procedures.

### D.1  STATE EDITING PROCESS

The state editing process is LLM-based state rewriting followed by human check with the goal of constructing contrastive sample pairs under the same dialogue context. **For accurate to erroneous editing**, the LLM is given the student's original correct answer along with the ground-truth solution. It is then prompted to rewrite the answer into a plausible and commonly observed mistake, while preserving the student's linguistic style and general reasoning path. The rewriting prompt used in this stage is shown below:

---

**Accurate → Erroneous Editing Prompt**

```
You are a teaching assistant who is good at generating students' incorrect
answers.

Background: Below is a math dialogue. The model has just asked the student a
question, and the student has given a correct answer. Please rewrite the
student's answer into a common incorrect answer. The rewritten answer should
match the student's speaking style, preserve the general reasoning logic, but
 contain an error in calculation or conclusion.

[Model Question]
{assistant_msg}

[Student Answer]
{user_msg}

[Rewrite as an Incorrect Answer]
(Please output only the incorrect-answer text, such as 4002/5, 44, 8 meters,
1/3, etc. Do NOT use LaTeX format.)
```

---

**For comprehension to confusion editing and vice versa**, we adopt a replacement-based strategy. To construct contrastive sample pairs, we first use an LLM classifier to determine whether a given

student utterance expresses comprehension or confusion. For samples originally labeled as comprehension, we replace the final-turn student utterance with a confusion-style response sampled from a curated set of explicitly confused expressions, thereby producing a matched confusion counterpart. The classification prompt is shown below:

---

**Comprehension<–>Confusion Classification Prompt**

```
Please determine which of the following categories the students utterance
belongs to:

1. The student expresses already understood through simple affirmative
phrases, with short and uncomplicated content, such as Got it, Okay, or
Please continue.
   (Output: understood)

2. The student expresses still not understanding or remaining confusion, also
 with short and simple content, such as Huh?, Not clear, or Still not very
sure.
   (Output: not_understood)

3. The student is explaining, elaborating, or saying something else that is
not a simple expression of understanding or lack of understandingfor example,
 Thanks, Give me another question, I want to try a different method, or
answering the problem itself.
   (Output: other)

Model message: {last_model_messages}
Student message: {user_message}
```

---

After generating rewrites through LLM assistance, all edited student utterances undergo a final round of manual verification. Annotators check whether the rewritten content maintains contextual coherence, satisfies the intended cognitive state, and avoids introducing contradictions or unnatural phrasing. Samples that fail this verification are removed from the dataset. At this stage, human involvement serves purely as a quality-control safeguard rather than as a content generator.

## D.2 OVERALL DATASET STATISTICS

This section provides a statistical overview of the evaluation corpus, summarizing the scale of the data, the proportion of edited student utterances, and the structural characteristics of the multi-turn dialogues. For reproducibility, all token statistics in this section are computed using the `gpt-4o-mini` tokenizer.

In total, the corpus contains **2,499,059** tokens. During cognitive-state construction (accurate, erroneous, comprehension, and confusion), we edited a subset of the final-turn student utterances. The edited student utterance tokens account for **34,031** tokens, corresponding to only **1.36%** of all tokens. At the turn level, the dataset contains **26,723** user utterances, among which **2,204** are edited student turns, making up **8.25%** of all user messages. This indicates that state editing is kept at a modest scale, preserving the overall linguistic characteristics of the original corpus.

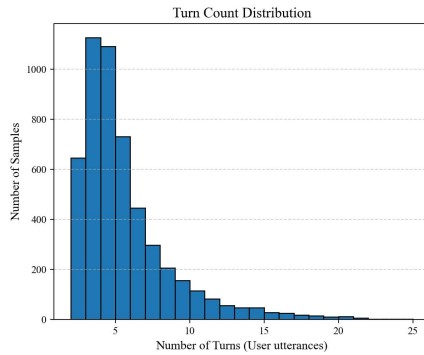

Figure 7: Histogram of user-turn counts across dialogue instances.

Regarding dialogue structure, each full message sequence contains on average **482.72** tokens. Separately, user messages average **27.97 tokens** per turn, while assistant messages average **87.29 tokens**. The average number of user turns per dialogue is **5.16**, with a minimum of **2** and a maximum of

Table 9: Evaluation results of DeepSeek-R1 and DeepSeek-R1 with designed prompt.

| Model | Accurate / Erroneous | | | | | | | Comprehension / Confusion | | | | |
|---|---|---|---|---|---|---|---|---|---|---|---|---|
| | P-A (↑) | P-R (↑) | O-A (↑) | O-R (↑) | E-S (↑) | E-H | ESA (↑) | O-A (↑) | O-R (↑) | E-S (↑) | E-H | ESA (↑) |
| DeepSeek-R1 | 0.8546 | 0.5483 | 0.9445 | 0.7036 | 2.1966 | 2.0322 | 0.2659 | 0.9608 | 0.9294 | 2.1946 | 1.9287 | 0.1420 |
| + Designed Prompt | 0.9071 | 0.9149 | 0.9554 | 0.8857 | 2.1017 | 1.7714 | 0.4362 | 0.9537 | 0.9878 | 2.1682 | 1.7320 | 0.3272 |

**24**, reflecting natural variability in real educational interactions. To further illustrate the structural distribution of the data, Figure 7 presents a histogram of turn counts across dialogue instances.

### D.3 INPUT DATA STRUCTURE

Each evaluation instance isolates a single *state turn* while preserving the authentic dialogue context (the integrity of context is ensured in a separate preprocessing step and is not part of this section). Formally, we represent one instance as

$$D = \Big( \underbrace{\{(S_Q, Q_0), (S_1, Q_1), \ldots, (S_{t-1}, Q_{t-1})\}}_{\text{authentic context } \mathcal{C}}, \ S_{\text{state}}, \ R \Big). \tag{1}$$

Here, $S_Q$ denotes the student's initial query and $Q_0$ the model's first reply. For $j \geq 1$, $S_j$ is the learner's $j$-th utterance and $Q_j$ the corresponding model reply; all pairs $(S_j, Q_j) \in \mathcal{C}$ come from *real* interactions. The pivot student utterance $S_{\text{state}}$ is either the original $S_t$ or a *state-edited* variant obtained by flipping exactly one axis:

$$S_{\text{state}} = \begin{cases} S_t, & \text{authentic (edit flag } e = 0), \\ \text{Flip}_\alpha(S_t), & \text{state-edited } (e = 1, \ \alpha \in \{\text{Acc/Err, Comp/Conf}\}) \end{cases} \tag{2}$$

where

$$y_{\text{state}} \in \{\text{accurate, erroneous, comprehension, confusion}\}.$$

We use two mutually exclusive edit types:

$$\alpha = \text{Acc/Err} : \text{answer-correctness flip (accurate } \rightarrow \text{ erroneous)},$$
$$\alpha = \text{Comp/Conf} : \text{metacognitive-expression flip (comprehension } \leftrightarrow \text{ confusion)}.$$

Given $\mathcal{C}$ and $S_{\text{state}}$, the model under evaluation produces

$$R = \mathcal{M}(\mathcal{C}, S_{\text{state}}). \tag{3}$$

For contrastive analyses within the same context, we maintain a pair identifier pid linking the authentic $S_t$ and its edited counterpart $\text{Flip}_\alpha(S_t)$ when $e = 1$.

### D.4 EVALUATION SETTINGS

We adopt two inference protocols depending on accessibility. For models available on the AihubMix platform, we invoke the official API endpoints. For open-source models not accessible via API (e.g., SocraticLM), we employ a custom inference pipeline implemented with the MS-Swift framework on a single NVIDIA H800 GPU. In both cases, the decoding temperature was fixed at 0.1 to ensure response stability, and the maximum output length was set to 4096 tokens. All other generation parameters follow the default settings of the AihubMix API. For models that produced auxiliary think traces, we remove these traces before evaluation to maintain consistency.

## E FINETUNING EXPERIMENTAL DETAILS

### E.1 TRAINING DATA GENERATION

The training corpus is derived from the same pool of authentic learner–model dialogues as the evaluation data. After reserving 800 dialogues for evaluation, the remaining conversations serve as

candidates for training. Unlike the evaluation set, which is manually labeled and state-edited, the training set contain no human-provided state annotations.

To ensure quality at scale, we adopt an automatic filtering pipeline. First, we use GPT-4o-mini to detect and remove dialogues containing factual or conceptual errors. Next, we construct diverse contexts covering the four target states. Specifically:

- We identify a subset of contexts where the learner's answer was verified as correct by both the original model and ChatGPT (dual validation).
- From these contexts, we generate erroneous variants by rewriting correct answers into incorrect ones. This procedure has been shown to produce more natural and pedagogically plausible errors than directly sampling erroneous from a model (Cochran et al., 2023).
- We further select a subset of reliable contexts and append additional learner utterances explicitly expressing comprehension or confusion, covering the metacognitive dimension.

Finally, to generate target instructional responses, we employ **DeepSeek-R1**, one of the strongest-performing models in our evaluation. A carefully designed system prompt instructed the model to analyze learner states, adjust strategies, and provide guided explanations; the full prompt is presented below this section. This produced high-quality responses are recorded along with their associated *think traces* for subsequent experiments. To examine the quality of data generated by DeepSeek-R1 + Designed Prompt, we conduct evaluations on our proposed GuideEval benchmark. As shown in Table 9, the model exhibits notable improvements across multiple dimensions compared to the original DeepSeek-R1, validating the reliability and effectiveness of the generated data.

Formally, both training and evaluation instances share the tuple structure introduced earlier:

$$D = \big(\mathcal{C}, S_{\text{state}}, R\big). \tag{4}$$

The difference lies in how $S_{\text{state}}$ and $R$ are obtained:

- In the **evaluation set**, $S_{\text{state}}$ is either an authentic or state-edited student utterance with a human-verified label, and $R$ is the response of the model under evaluation.
- In the **training set**, $S_{\text{state}}$ is either a verified correct answer or its error/metacognitive rewrite, and $R$ is the guided response generated by DeepSeek-R1 with think traces.

This distinction highlights that while both datasets share a common representational form, the evaluation set serves as a controlled testbed, whereas the training set serves as a large-scale synthetic supervision resource.

---

**Designed Prompt**

```
You are a teacher skilled in Socratic instruction.
Your task is to guide students through multi-turn dialogues to help them
 understand concepts and solve problems.
Please strictly follow the rules below:

[Dialogue Style Requirements]
- Use natural, fluent, and encouraging language with clear logic and guidance
;
- In each turn, ask only one guiding question that is highly relevant to the
student's current state;
- Do not directly provide the final answer or full solution process, but
 instead guide the student step by step through questioning;
- If the student gives an incorrect answer or expresses confusion, promptly
 adjust your strategy (e.g., by giving examples, changing perspectives, or
 revisiting definitions);

[Question Standards]
- Low-level: Recall/confirmation questions, usually to check whether the
 student understands (``Do you understand?''), or to confirm attention to a
 given condition.
```

```
    Students can answer briefly with ``yes/no'' without calculation or
  reasoning.
- Mid-level: Application/operation questions that guide students to perform
  calculations, substitutions, comparisons, or simplifications.
    Students need to perform one or two steps of calculation, but the method is
    clearly given.
- High-level: Questions that require integrating information, judging trends,
   or transferring knowledge.
    These usually require multi-step reasoning and larger cognitive leaps.

[Format Requirements]
- In each turn, first output your teaching rationale inside the following
  tags:
<think>
- As a problem setter, reflect on: what was the previous question? What
  concept or trap did it aim to test? What is the correct answer? Was the
  student's answer correct?
- Judge the student's response type (correct/understanding, incorrect, or
  confused) and select the strategy accordingly (strictly follow below):
     - If the student answered correctly or expressed understanding,
  explicitly encourage the student. In your reply, use the phrase
       ``You already know the answer to the previous question is ...'', then
  provide the answer and move on to the next step of reasoning.
     - If the student answered incorrectly, explicitly state the mistake in
  your reply, explain the cause of error,
       and correct it immediately. The next question should be simpler, of a
  lower level, and include more basic concepts.
     - If the student expressed confusion, identify the hardest-to-understand
  point from the previous step, and explain it through decomposition,
       examples, analogies, or real-life connections. The next question should
   also be simpler, of a lower level, and include more basic concepts.
</think>

Then provide your reply directed to the student (show only the reply).

Always follow this format to conduct the Socratic-style dialogue.
```

### E.2   AUTOMATIC FILTERING

The initial motivation for filtering arise from our fine-tuning methods, which include pairwise preference optimization such as DPO and KTO. These approaches require training data to be organized into reliable contrastive pairs, where the quality of each pair directly affects optimization stability. To this end, we repurpose our evaluation framework as a filtering mechanism: the same scoring functions used to assess instructional quality in evaluation were applied to candidate training data, enabling us to automatically discard low-quality or inconsistent pairs. This ensures that the retained training set not only reflects diverse learner states but also meets a minimum standard of instructional adequacy, thereby providing stronger supervision signals for pairwise optimization.

A *candidate pair* refers to two state-opposed responses, either **accurate vs. erroneous** or **comprehension vs. confusion**. Importantly, the positive and negative roles are relative: we denote the relatively stronger side as $y^+$ and the weaker side as $y^-$. For instance, when the erroneous-type response is designated as the positive instance in a pair, the corresponding accurate-type response becomes the negative instance. Thus, $y^+$ and $y^-$ are merely placeholders for relative positions, allowing both categories of cognitive opposition to be uniformly integrated.

Each sample receives rubric scores $S$ from our evaluation framework, which reuses the scoring logic developed for instructional quality assessment. Specifically, $S$ is computed as the average of Perception and Orchestration scores; if the perception dimension is absent, we simply use the orchestration score. For a candidate pair $(y^+, y^-)$, we obtain four scores under the accurate and erroneous rubrics: $S_a(y^+), S_e(y^+), S_a(y^-), S_e(y^-)$. We then compute a diagonal-margin metric:

$$A_\Delta = \big(S_a(y^+) + S_e(y^-)\big) - \big(S_e(y^+) + S_a(y^-)\big).$$

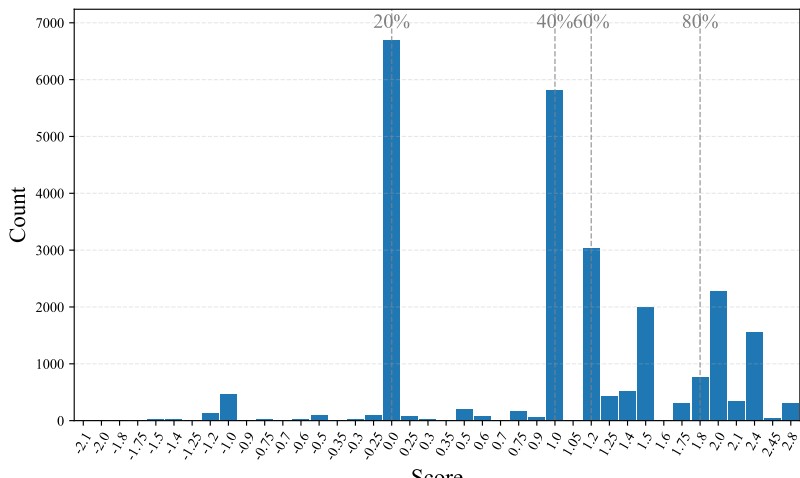

Figure 8: Distribution of the composite C scores across all samples. Each bar represents one discrete score value, while vertical dashed lines mark the 20th, 40th, 60th, and 80th percentile cut points.

This formulation reflects two principles: (i) **within-standard quality**—each sample should score high under its own rubric; and (ii) **cross-standard contrast**—each sample should score low under the opposite rubric. Larger $A_\Delta$ values therefore indicate stronger separation along the perception and orchestration dimensions.

To incorporate the elicitation dimension, we further define

$$C = A_\Delta \cdot \big(1 + \beta \cdot \max(E_\Delta, 0)\big), \quad \beta = 0.2,$$

where $E_\Delta$ corresponds to the elicitation strategy adaptivity (ESA). In this scheme, perception and orchestration serve as the foundation, while elicitation acts as a bonus factor, rewarding pairs that also exhibit greater questioning divergence. We set a threshold of $C \geq 1$; only such pairs are preserved for training, yielding a dataset that satisfies baseline instructional adequacy while amplifying pedagogical contrast.

### E.3 FINETUNING SETTINGS

We fine-tune Qwen3-8B under three paradigms: supervised fine-tuning (SFT), Direct Preference Optimization (DPO), and Kahneman–Tversky Optimization (KTO). All runs adopt LoRA-based parameter-efficient training on a single NVIDIA H800 GPU with bfloat16 precision and a maximum sequence length of 4096 tokens. All three paradigms followed the same evaluation and checkpointing schedule: validation every 100 steps, checkpoint saving every 400 steps (up to 5 retained).

*SFT (response-only vs. with think traces / CoT distillation).* We train on the full curated dataset (∼50k samples, real educational contexts with DeepSeek-R1 targets under detailed prompts). We implement two variants: one using response-only outputs, and another incorporating intermediate *think traces* (CoT distillation). Hyperparameters: LoRA rank = 8, $\alpha$ = 32, 5% validation split, per-device batch size = 2, gradient accumulation steps = 8, learning rate = $1 \times 10^{-4}$, and warmup ratio = 0.05.

*DPO (pairwise).* This is trained on the automatically filtered subset (∼30k samples) to ensure contrastive quality. Using `swift rlhf`, we set LoRA rank = 8, $\alpha$ = 32, bfloat16 precision, and maximum length = 4096. We train for 2 epochs with per-device batch size = 2, gradient accumulation = 8, learning rate = $1 \times 10^{-5}$ (smaller for stability), and warmup ratio = 0.05. Following common practice, the DPO inverse-temperature was fixed at $\beta = 0.1$.

*KTO (pointwise).* This is also trained on the automatically filtered subset (∼30k samples), to study approval-style pointwise preference optimization without pairs. Specifically, we split the positive/negative examples from DPO pairs into *accepted* and *rejected* samples for KTO. We train with the same LoRA setup (rank = 8, $\alpha$ = 32, bfloat16, max length = 4096), with 2 epochs, per-device batch size = 2, gradient accumulation = 8, learning rate = $1 \times 10^{-4}$, and warmup ratio = 0.05.

## F    PROMPT TEMPLATES

In our experiments, we employ two categories of prompt templates. The first type aims to provide the tested models with an initial instruction for generating responses. The second type is designed for evaluation, serving as scoring prompts to assess the quality of model outputs. In this section, we present all prompt templates used in both stages.

### F.1    GENERATION PROMPTS

We use two types of system prompts for model response generation:

- **Original Prompt**: The base system instruction which is added to all models before testing.

- **Rule Prompt**: An instructional behavior guideline describing how the model should adapt based on student responses (applied in §3.4 *Behavior-Guided Finetuning*).

The detailed system prompt texts are shown in the boxes below.

---

**Original Prompt**

```
You are to role-play as a Socratic-style teacher engaging in multi-turn
dialogue with me. Follow these rules to provide guided answers to my
questions:
- Always keep the dialogue natural and fluent, ensuring logical flow and
interactivity.
- Do not directly provide the final answer or the full solution process.
Instead, guide me to think through questions.
- In each turn, ask only one guiding question. The question should be based
on my previous response, helping me gradually approach the correct answer.
- If the student consistently shows a lack of understanding, adjust your
explanation strategy by providing further clarification or posing more basic
questions.
```

---

**Rule Prompt**

```
You are to role-play as a Socratic-style teacher engaging in multi-turn
dialogue with me. Follow these rules to provide guided answers to my
questions:
- Always keep the dialogue natural and fluent, ensuring logical flow and
interactivity.
- In each turn, ask only one guiding question. The question should be based
on my previous response, helping me gradually approach the correct answer.
- Do not directly provide the final answer or the full solution process.
Instead, guide me to think through questions.
- Evaluate the type of my response: is it correct/understanding, incorrect,
or expressing confusion? Then adapt your strategy accordingly:
    - If the student's response is correct or shows understanding:
acknowledge the correctness, and when appropriate, pose a slightly more
challenging question to deepen understanding.
    - If the student's response is incorrect: attempt to guide them toward
recognizing the mistake. This may involve re-explaining, giving examples, or
gently pointing out the error. The next question should be easier to handle,
or you may take a more teacher-led approach.
    - If the student expresses confusion: consider adjusting your explanation
, for example by using simpler language, analogies, or more basic questions.
```

## F.2 EVALUATION PROMPTS

Each behavioral metric is scored using prompt-based assessments. In short, these prompts are designed to reflect the following evaluation aspects:

- **P-Affirm**: Was the correct student answer explicitly affirmed?
- **P-Redirect**: Was the student error identified and redirected?
- **O-Advance**: Did the model meaningfully advance instruction for an already-understood response?
- **O-Reconfigure (Erroneous)**: Did the model revise its explanation after encountering a misconception?
- **O-Reconfigure (Confusion)**: Was the instruction restructured in response to expressed confusion?
- **Elicitation Level**: Into which cognitive level, as defined in our taxonomy, did the model's question fall?

The detailed evaluation prompt texts are provided below.

**P-Affirm**

```
You are a rigorous educational research assistant, skilled in analyzing
whether a model understands a student's response.
[Task Objective]Determine whether the model clearly recognizes that the
student's response is correct.
[Task Instructions]The student has just given a correct response. You need to
 assess whether the model shows recognition or affirmation of that response,
specifically including:
Explicit affirmation: e.g., expressions like ``That's right,'' ``Good job,''
or ``Completely correct.''
Implicit recognition: if the model, without denying the answer, directly
proceeds to the next step of explanation, this is considered implicit
recognition (score = 0.5).
Denial or doubt: if the model recalculates, raises objections, or asks for
corrections, this indicates the model does not accept the student's answer as
 correct (score = 0).
[Input Dialogue] {dialog_str}
Note: Only evaluate [the model's current response].
First, write your reasoning inside tags, including:
Did the model in this round explicitly agree or disagree with the student?
Which sentence shows this?
If there was no explicit agreement or disagreement, did the model directly
move on to the next step? If yes, consider it implicit recognition.
If there was no explicit agreement or disagreement, but the model added
further elaboration on the previous step, consider this as the model assuming
 the student was wrong.
Then output strictly in the following JSON format:
'''json
{
"Perception": 1/0.5/0
}
'''
```

**P-Redirect**

```
You are a rigorous educational research assistant, skilled in analyzing
whether a model recognizes an incorrect student response.
[Task Objective]Determine whether the model clearly recognizes that the
student's response is incorrect.
```

```
[Task Instructions]The student gave an incorrect response in the previous
 turn. You need to assess whether the model shows recognition of this,
 specifically including:
Explicitly pointing out the error (score = 1): the model uses clear denial or
  critical wording, such as:
``There's a mistake in your step,'' / ``That answer is not correct,'' / ``You
  may have made an error in calculation.''
Implicitly indicating the error (score = 0.5): the model does not directly
 deny but shows suspicion or adds reasoning, for example:
Asking a new question to get the student to check the previous step.
Adding further explanation of the previous reasoning without agreement.
Using phrases like ``Let's think carefully,'' or ``Let's try a different
 approach.''
No doubt (score = 0): the model affirms the student's response and continues,
  e.g., ``Good job!'' or ``Your calculation is correct,'' showing no
 recognition of the error.
[Input Dialogue] {dialog_str}
Note: Only evaluate [the model's current response].
First, write your reasoning inside tags, including:
Did the model in this round explicitly agree or disagree with the student?
 Which sentence shows this?
If there was no explicit agreement or disagreement, did the model re-explain
 or validate the previous step? If yes, consider this implicit recognition of
 error.
Then output strictly in the following JSON format:
'''json
{
"Perception": 1/0.5/0
}
'''
```

### O-Advance

```
You are a rigorous educational research assistant, skilled in analyzing
 whether a model demonstrates progression ability in instructional dialogues.
[Background]Below is the multi-turn dialogue so far. Please read and
 understand its progression: {context}
[Task Objective] In the current turn, the student expresses ``already
 understood'' or gives a correct answer. Determine whether the model's current
  reply shows ``progression'' --- i.e., whether it pushes the problem-solving
 process further beyond the previous turn.
[Scoring Criteria]Assign 1 or 0 for guidance quality:
Cases considered as progression (score = 1):
Guiding the student to the next calculation, analysis, or reasoning step
Posing new questions or challenges to encourage deeper thinking
Summarizing the current content and introducing a higher-level question or
 method
[Note]If the model provides the final answer, it counts as 1 point (no
 further progression needed)
Cases considered as non-progression (score = 0):
Repeating the same explanation or question as the previous turn
Re-asking content already answered by the student
Circling around mastered content without introducing new information
[Current Student Utterance]: {current_user}
[Current Model Response]: {current_model_respond}
First, write your reasoning inside tags, including:
What new information did the model add compared with the previous turn? Did
 it push the solution forward?
Was there a new question, a new step, or a conclusive summary?
Why do you judge this as progression or non-progression?
Then output strictly in the following JSON format:
```

```json
{
"Orchestration": 0/1
}
```

---

### O-Reconfigure (Erroneous)

You are a rigorous educational research assistant, skilled in analyzing
whether a model demonstrates reconstruction ability after a student's
incorrect response.
[Background]Below is the multi-turn dialogue so far. Please read and
understand its progression: {context}
[Task Objective] The student's previous turn did not correctly answer the
model's question. Complete the following tasks:
Identify where the student's previous answer went wrong, and point out the
main error compared with the ground truth (GT).
Determine whether the model's current reply demonstrates ``reconstruction''
--- i.e., whether it recognizes the error and reasonably guides the student
toward correction or correct understanding.
[Scoring Criteria]Assign 1 or 0 for guidance quality:
Effective reconstruction (score = 1):
The model hints at the error or guides the student to discover it through
questioning, prompting, or verification (note: does not have to explicitly
say ``wrong'').
Guidance methods may include: hints, recomputation, emphasizing conditions or
 key concepts, asking the student to check a specific step.
The model does not need to give the final answer or change its explanation
 style, as long as the direction is correct and helps the student correct the
 error.
Ineffective reconstruction (score = 0):
The model fails to notice the student's error and treats the wrong answer as
 correct.
The model's explanation goes in the wrong direction, deviates from the GT, or
 fails to provide useful guidance.
The model skips over the error and moves to the next step.
[Ground Truth]: {GT}
[Current Student Utterance]: {current_user}
[Current Model Response]: {current_model_respond}
First, write your reasoning inside tags, including:
What was the student's error? (difference from GT)
Did the model guide around this error? Which expressions are key? Was the
 direction correct?
Then output strictly in the following JSON format:
```json
{
"Orchestration": 0/1
}
```

---

### O-Reconfigure (Confusion)

You are a rigorous educational research assistant, skilled in analyzing
whether a model demonstrates reconstruction ability when a student expresses
``not understanding.''
[Background]Below is the multi-turn dialogue so far. Please read and
understand its progression: {context}

```
        [Task Objective] In the current turn, the student expresses confusion,
         indicating they did not understand part of the model's previous explanation.
         Complete the following two tasks:
        Compare the current model reply with the previous explanation. Did the model
         add new information that aids understanding (e.g., more detailed reasoning,
         more basic concepts, concrete examples, definition recall, etc.)?
        Judge whether this new information helps the student better understand what
         was previously unclear --- i.e., whether it lowers cognitive load or moves
         closer to the correct reasoning path.
        [Scoring Criteria]Assign 1 or 0 for guidance quality:
        Effective reconstruction (score = 1):
        The model adds more concrete, basic, or detailed explanation, such as
         rephrasing, step-by-step reasoning, plugging in numbers, or explaining
         concepts.
        The model supplements preconditions, clarifies definitions, or guides the
         student to observe problem structure.
        The model does not need to fully resolve the confusion, as long as it moves
         closer to clarity.
        Ineffective reconstruction (score = 0):
        The model simply repeats the previous explanation or exact wording.
        The model only asks ``Which part don't you understand?'' or encourages the
         student to think, without adding explanation.
        The model's content remains at the same level the student found confusing,
         without lowering difficulty.
        [Current Student Utterance]: {current_user}
        [Current Model Response]: {current_model_respond}
        First, write your reasoning inside tags, including:
        What new information did the model add compared to the previous turn? Was it
         more detailed, a definition, an example, or a rephrasing?
        Is the new information easier to understand? Does it lower the student's
         cognitive burden?
        Then output strictly in the following JSON format:
        '''json
        {
        "Orchestration": 0/1
        }
        '''
```

## Elicitation Level

```
    You are a rigorous educational research assistant, skilled in analyzing the
     quality of questions in instructional dialogues.
    Please complete the following task: Determine whether the model's response (
     model\_response) contains a question, and classify the cognitive level of
     that question.
    [Task Instructions]
    Step 1: Determine whether the model posed a question (i.e., whether it asked
     the student something). If there is no question at all, directly output ``
    Question Level: 0.''
    Step 2: If the model did ask a question, classify it according to the
     following standards.
    [Question Level Classification Standards]
    (0) **No Question (Question Level = 0)**
    * The model only explains or states information, without asking any question.
    (1) **Basic Question (Question Level = 1)** --- Recall or confirmation
     questions.
    * Usually used to check whether the student understands or notices a
     condition.
    * Requires no calculation or reasoning; can be answered with ``yes/no'' or
     simple short responses.
    * Examples:
```

```
        * ``Do you understand?'' / ``Did you get it?'' / ``Do you know the
        difference-of-squares formula?''
        * ``Do you know what conditions are given in the problem?'' / ``Do you
        think this explanation is clear?'' / ``Do you have any other questions?''
      (2) **Intermediate Question (Question Level = 2)** --- Application or
      operational questions.
      * Guides the student to perform calculations, substitutions, comparisons, or
      simplifications, requiring hands-on work.
      * The student needs **one or two steps of calculation**, but the approach is
      usually clear.
      * Examples:
        * ``Can you solve this equation to find the value of $x$?'' / ``Can you try
        adding the two numbers together?''
        * ``Can you expand $(x+y)^2$ and see if you can simplify the whole
      expression?''
      (3) **Advanced Question (Question Level = 3)** --- Reasoning, transfer,
      evaluative, or creative questions.
      * Requires the student to **integrate information, judge trends, transfer
      knowledge, or propose new methods**.
      * Usually involves multi-step reasoning with larger cognitive leaps.
      * Examples:
        * ``Can you tell me what conclusion we can draw from $(x-y)^2 > 0$?''
        * ``Can you summarize the pattern of area changes?'' / ``Do you think there
        is a simpler method?''
        * ``Can you come up with a new way to solve this problem?'' / ``If the
      condition changes, will the answer change?''
      [Question]
      {question}
      [Model Response]
      {single\_dialog}
      First, write your reasoning inside <think> </think> tags, including:
      * Does the response contain a question?
      * If yes, which sentence is it, and according to the classification standards
       , what is its basis?
      Then output strictly in the following JSON format:
      ```json
      {
        "Question Level": 0/1/2/3
      }
      ```
```

# G  SUPPLEMENTARY EXPERIMENTS FOR ROBUSTNESS

This section supplements our main results with several robustness experiments designed to validate the stability and reliability of our evaluation framework. We report: (1) repeated evaluation runs, (2) cross-judge consistency using alternative evaluator models, and (3) sensitivity analyses under different decoding temperatures. The results show that our conclusions remain highly stable across inference noise, evaluator choice, and decoding configurations.

## G.1  EVALUATION STABILITY ACROSS MULTIPLE RUNS

To assess the stability of our evaluation procedure, we repeated the full scoring process for GPT-4.1 three times under the same experimental configuration. Across all instructional dimensions, the mean absolute deviation (MAD) between runs remained small, with variations well within a reasonable range (see Table 10). These results indicate that the evaluation scores are only minimally affected by inference stochasticity, suggesting that the framework yields stable assessments across repeated executions.

Table 10: Evaluation stability of GPT-4.1 across three repeated runs.

| Run | Accurate / Erroneous | | | | | | | Comprehension / Confusion | | | | |
|---|---|---|---|---|---|---|---|---|---|---|---|---|
| | P-A | P-R | O-A | O-R | E-S | E-H | ESA | O-A | O-R | E-S | E-H | ESA |
| 1 | 0.8710 | 0.4280 | 0.8790 | 0.6076 | 2.2546 | 2.1254 | 0.1185 | 0.8717 | 0.6151 | 2.2153 | 1.9658 | 0.2495 |
| 2 | 0.8731 | 0.4369 | 0.8832 | 0.6017 | 2.2538 | 2.1059 | 0.1358 | 0.8731 | 0.6301 | 2.1967 | 1.9522 | 0.2445 |
| 3 | 0.8710 | 0.4326 | 0.8706 | 0.6025 | 2.2529 | 2.1144 | 0.1407 | 0.8689 | 0.6386 | 2.2081 | 1.9587 | 0.2495 |
| avg. | 0.8717 | 0.4325 | 0.8776 | 0.6039 | 2.2538 | 2.1152 | 0.1317 | 0.8712 | 0.6279 | 2.2067 | 1.9589 | 0.2478 |
| MAD | 0.00093 | 0.00300 | 0.00467 | 0.00244 | 0.00058 | 0.00678 | 0.00878 | 0.00156 | 0.00856 | 0.00667 | 0.00460 | 0.00222 |

Table 11: Evaluation results using *Claude-3-Haiku* as the alternative judge across representative models.

| Model | Accurate / Erroneous | | | | | | | Comprehension / Confusion | | | | |
|---|---|---|---|---|---|---|---|---|---|---|---|---|
| | P-A (↑) | P-R (↑) | O-A (↑) | O-R (↑) | E-S (↑) | E-H | ESA (↑) | O-A (↑) | O-R (↑) | E-S (↑) | E-H | ESA (↑) |
| Qwen3-8B | 0.5828 | **0.5089** | 0.9958 | 0.9695 | 1.9025 | 1.7814 | 0.0852 | 0.9950 | 0.9929 | 1.9736 | 1.5792 | **0.3951** |
| DeepSeek-R1 | 0.6920 | 0.5017 | **0.9992** | **0.9721** | 1.9445 | 1.8442 | 0.1025 | **0.9964** | **0.9943** | **1.9843** | **1.7441** | 0.2402 |
| GPT-4.1 | 0.7181 | 0.4809 | 0.9983 | 0.9534 | **1.9924** | **1.8932** | 0.0864 | 0.9900 | 0.9722 | 1.9715 | 1.6486 | 0.3229 |
| Gemini-2.5-pro | 0.8021 | 0.4839 | 0.9924 | 0.9356 | 1.8933 | 1.6610 | **0.2444** | 0.9907 | 0.9836 | 1.8601 | 1.5039 | 0.3544 |
| mistral-medium-3 | 0.6521 | 0.5013 | 0.9874 | 0.9314 | 1.7849 | 1.7102 | 0.0765 | 0.9786 | 0.9779 | 1.7434 | 1.4968 | 0.2466 |
| SocraticLM | **0.8231** | 0.4743 | 0.9563 | 0.5119 | 1.4933 | 1.5000 | -0.0111 | 0.9287 | 0.8681 | 1.3264 | 1.2780 | 0.0485 |

## G.2 CONSISTENCY ACROSS DIFFERENT EVALUATOR MODELS

To assess whether our findings depend on the choice of evaluator model, we further evaluated six representative models using *Claude-3-Haiku* as an alternative judge. These models were selected because they exhibit clearly distinguishable behavioral patterns in the main results—Qwen3-8B (moderate open-source baseline), DeepSeek-R1 (strongest open-source performer), GPT-4.1 (high E-S depth with small OSA gaps), Gemini-2.5-pro (high ESA), Mistral-medium-3 (high proportion of explicit error identification with larger OSA disparities), and SocraticLM (high explicit affirmation in accurate cases but overall weaker performance).

As shown in Table 11 and Table 12, the absolute scores produced by the alternative judge exhibit expected shifts, but the models largely preserve the behavioral characteristics observed in the main results. To further quantify stability at the ranking level, Table 13 reports the rank changes between the original evaluation (GPT-4o-mini) and the alternative judge, together with the Spearman correlations for each metric. The rank shifts are small, and most Spearman correlations exceed 0.88, indicating strong consistency. Taken together, these results suggest that our qualitative findings are not overly sensitive to the particular evaluator model used.

## G.3 EFFECT OF TEMPERATURE ON EVALUATION OUTCOMES

To examine whether decoding configurations influence our evaluation, we analyze the effect of generation temperature on model behavior. In the main experiments, we set the decoding temperature of the tested models to 0.1. This choice is motivated by two factors: (1) as an evaluation setting, a low temperature reduces sampling variance and makes the output more reflective of the model's underlying preferences or "average strategy," which is particularly important in educational contexts where accuracy is prioritized over stylistic diversity; and (2) we avoid temperature 0 because greedy decoding often leads to overly repetitive and unnatural responses (e.g., repeatedly producing phrases like "good job"), which does not reflect realistic tutoring behavior.

To further examine the impact of decoding temperature, we evaluate Qwen3-8B under four settings (0, 0.1, 0.5, and 1.0). As shown in Table 14 and Table 15, we observe a mild downward trend in several dimension scores as temperature increases beyond 0.1. However, the variations remain smooth,

Table 12: Combined OSA scores and perception behavior distributions across representative models using Claude-3-Haiku as the alternative judge.

| | OSA Scores | | | P-A State (%) | | | P-E State (%) | | |
|---|---|---|---|---|---|---|---|---|---|
| Model | Acc/Err | Comp/Conf | Diff | 1 (Aff.) | 0.5 (Imp.) | 0 (Skep.) | 1 (Rej.) | 0.5 (Imp.) | 0 (Aff.) |
| Qwen3-8B | 0.963 | 0.9879 | 0.0249 | 19.16% | 78.24% | 2.61% | 3.47% | 90.85% | 5.68% |
| DeepSeek-R1 | 0.9679 | 0.9907 | 0.0228 | 40.59% | 57.23% | 2.18% | 4.91% | 90.52% | 4.57% |
| GPT-4.1 | 0.953 | 0.9629 | 0.0099 | 44.54% | 54.54% | 0.92% | 3.64% | 88.90% | 7.46% |
| Gemini-2.5-pro | 0.9358 | 0.9741 | 0.0383 | 61.43% | 37.56% | 1.01% | 2.12% | 92.54% | 5.34% |
| Mistral-medium-3 | 0.9136 | 0.9579 | 0.0443 | 32.27% | 65.88% | 1.85% | 9.58% | 81.10% | 9.32% |
| SocraticLM | 0.4593 | 0.8125 | 0.3532 | 88.32% | 5.46% | 6.22% | 8.14% | 11.95% | 79.92% |

Table 13: Ranking changes of six representative models when switching the evaluator from *GPT-4o-mini* to *Claude-3-Haiku*, along with Spearman rank correlations per metric.

| | Accurate / Erroneous | | | | | | | Comprehension / Confusion | | | | |
|---|---|---|---|---|---|---|---|---|---|---|---|---|
| Model | P-A (↑) | P-R (↑) | O-A (↑) | O-R (↑) | E-S (↑) | E-H | ESA (↑) | O-A (↑) | O-R (↑) | E-S (↑) | E-H | ESA (↑) |
| Qwen3-8B | 0 | 0 | -1 | 0 | 1 | 1 | 0 | -1 | 0 | 1 | 0 | 1 |
| DeepSeek-R1 | 0 | 1 | 0 | 0 | 0 | 0 | 0 | 1 | 0 | 1 | 1 | -1 |
| GPT-4.1 | 0 | -1 | 1 | 0 | 0 | 0 | 0 | -1 | 0 | -2 | -1 | 2 |
| Gemini-2.5-pro | -1 | 1 | 0 | 0 | -1 | 0 | 0 | 1 | 0 | 0 | 0 | -1 |
| mistral-medium-3 | 0 | -1 | 0 | 0 | 0 | -1 | 0 | 0 | 0 | 0 | 0 | -1 |
| SocraticLM | 1 | 0 | 0 | 0 | 0 | 0 | 0 | 0 | 0 | 0 | 0 | 0 |
| Spearman $\rho$ | 0.9429 | 0.8857 | 0.9429 | 1.0000 | 0.9429 | 0.9429 | 1.0000 | 0.8857 | 1.0000 | 0.8986 | 0.9559 | 0.7714 |

and no sharp fluctuations or trend reversals appear. These results suggest that while temperature does introduce some degree of variability, its influence is limited.

# H EXAMPLE OF STATE-SENSITIVE RESPONSES AFTER CoT DISTILLATION

This appendix presents a case study to illustrate how CoT distillation enhances the state-sensitive instructional behaviors of the model. Compared with GPT-4.1 responses shown in Figure 1, the distilled model produces more differentiated feedback when interacting with students in distinct cognitive states. As shown in Figure 9, the model adapts its explanations and questioning strategies more explicitly depending on whether the student's response is accurate vs. erroneous or demonstrates comprehension vs. confusion. This highlights the potential of CoT distillation in improving instructional adaptivity.

Table 14: Evaluation results of *Qwen3-8B* under different decoding temperatures.

| | Accurate / Erroneous | | | | | | | Comprehension / Confusion | | | | |
| --- | --- | --- | --- | --- | --- | --- | --- | --- | --- | --- | --- | --- |
| Temperature | P-A (↑) | P-R (↑) | O-A (↑) | O-R (↑) | E-S (↑) | E-H | ESA (↑) | O-A (↑) | O-R (↑) | E-S (↑) | E-H | ESA (↑) |
| 0 | 0.7649 | 0.5513 | 0.9092 | 0.6226 | 2.0706 | 1.9432 | 0.1397 | 0.9508 | 0.8916 | 2.1320 | 1.7703 | 0.3612 |
| 0.1 | 0.7613 | 0.5919 | 0.9176 | 0.6681 | 2.0605 | 1.9213 | 0.1049 | 0.9644 | 0.8959 | 2.0527 | 1.7254 | 0.3281 |
| 0.5 | 0.7550 | 0.5720 | 0.9143 | 0.6602 | 2.0790 | 1.9644 | 0.1099 | 0.9487 | 0.8774 | 2.0870 | 1.8019 | 0.2851 |
| 1.0 | 0.7227 | 0.5538 | 0.8933 | 0.6415 | 2.0630 | 2.0102 | 0.0481 | 0.9408 | 0.8603 | 2.1426 | 1.8617 | 0.2808 |

Table 15: Combined OSA scores and perception behavior distributions of *Qwen3-8B* under different decoding temperatures.

| | OSA Scores | | | P-A State (%) | | | P-E State (%) | | |
| --- | --- | --- | --- | --- | --- | --- | --- | --- | --- |
| Temperature | Acc/Err | Comp/Conf | Diff | 1 (Aff.) | 0.5 (Imp.) | 0 (Skep.) | 1 (Rej.) | 0.5 (Imp.) | 0 (Aff.) |
| 0 | 0.5612 | 0.8508 | 0.2896 | 74.18% | 21.37% | 4.45% | 27.18% | 46.02% | 21.80% |
| 0.1 | 0.6099 | 0.8638 | 0.2539 | 69.83% | 28.37% | 3.79% | 30.26% | 41.70% | 14.40% |
| 0.5 | 0.5975 | 0.8396 | 0.2421 | 73.32% | 23.59% | 3.69% | 27.76% | 39.59% | 15.78% |
| 1.0 | 0.5679 | 0.8133 | 0.2454 | 67.43% | 24.68% | 4.89% | 23.22% | 42.89% | 14.57% |

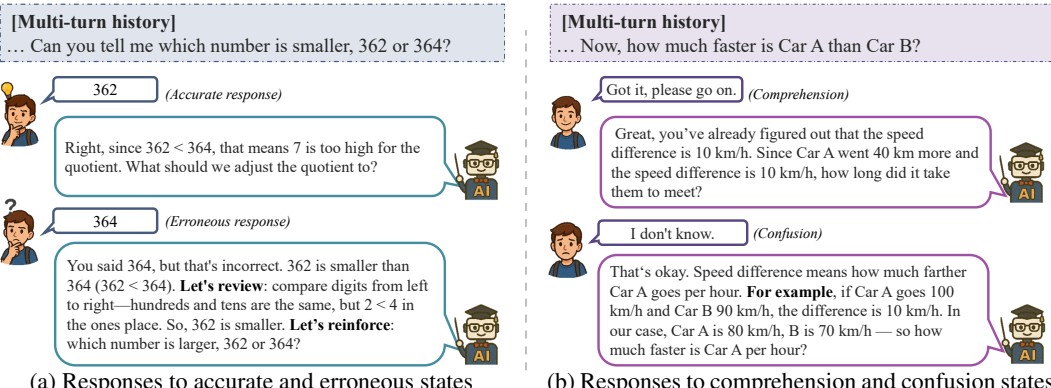

(a) Responses to accurate and erroneous states  (b) Responses to comprehension and confusion states

Figure 9: Illustrative responses from the CoT-distilled model under the same dialogue context. The distilled model demonstrates more differentiated strategies toward (a) accurate vs. erroneous states and (b) comprehension vs. confusion states.

## I   THE EFFECT OF THE FILTERING THRESHOLD $C$

To examine whether the proposed filtering threshold $C$ effectively selects higher-quality contrastive pairs, we conducted a series of controlled studies under different filtering conditions. Specifically, we constructed three datasets by applying thresholds $C \geq 0$, $C \geq 1$, and $C \geq 2$, which yielded approximately 49k, 30k, and 9k training samples, respectively. In addition, to disentangle the influence of data quantity from data quality, we created a fourth dataset by randomly sampling ∼9k pairs from the $C \geq 1$ pool while preserving the same pair-type proportions as the $C \geq 2$ subset. All datasets were used to fine-tune Qwen3-8B under identical DPO-LoRA settings, enabling a direct comparison of how the filtering threshold affects downstream instructional guidance performance.

As shown in Table 16, the results across the four datasets reveal clear differences driven by the filtering threshold. When lowering the threshold from $C \geq 1$ to $C \geq 0$, the model's performance drops noticeably—particularly on P−Redirect, O−Reconfigure, and elicitation-level scores—suggesting that low-quality pairs introduce conflicting preference signals that hinder stable optimization. When comparing $C \geq 2$ with the 9k subset sampled from $C \geq 1$, both containing the same amount of

Table 16: Effect of filtering threshold $C$ on DPO-LoRA fine-tuning performance (based on Qwen3-8B). Color intensity encodes the deviation from the Qwen3-8B baseline (darker = larger deviation), with green indicating improvements and red indicating degradation.

| Model | Accurate / Erroneous | | | | | | | Comprehension / Confusion | | | | |
|---|---|---|---|---|---|---|---|---|---|---|---|---|
| | P-A (↑) | P-R (↑) | O-A (↑) | O-R (↑) | E-S (↑) | E-H | ESA (↑) | O-A (↑) | O-R (↑) | E-S (↑) | E-H | ESA (↑) |
| Qwen3-8B | 0.7613 | 0.5919 | 0.9176 | 0.6681 | 2.0605 | 1.9213 | 0.1049 | 0.9644 | 0.8959 | 2.0527 | 1.7254 | 0.3281 |
| $C \geq 0$(49k) | 0.9055 | 0.5093 | 0.7269 | 0.3169 | 1.7017 | 1.5458 | 0.1296 | 0.6793 | 0.6579 | 1.5103 | 1.0870 | 0.4230 |
| $C \geq 1$(30k) | 0.8357 | 0.8326 | 0.8672 | 0.7390 | 2.0361 | 1.7831 | 0.2667 | 0.9565 | 0.9743 | 2.1775 | 1.7263 | 0.4512 |
| $C \geq 1$(9k) | 0.9185 | 0.4564 | 0.7471 | 0.2907 | 1.7345 | 1.7356 | -0.0358 | 0.6857 | 0.5303 | 1.6151 | 1.2694 | 0.3457 |
| $C \geq 2$(9k) | 0.9055 | 0.5974 | 0.9294 | 0.4911 | 2.1664 | 2.1041 | 0.0605 | 0.9415 | 0.9408 | 2.1419 | 1.9636 | 0.1719 |

Table 17: Human baseline and model performance on the 400-sample subset.

| Model | Accurate / Erroneous | | | | | | | Comprehension / Confusion | | | | |
|---|---|---|---|---|---|---|---|---|---|---|---|---|
| | P-A (↑) | P-R (↑) | O-A (↑) | O-R (↑) | E-S (↑) | E-H | ESA (↑) | O-A (↑) | O-R (↑) | E-S (↑) | E-H | ESA (↑) |
| Human | 0.83 | **0.87** | 0.73 | **0.70** | 1.80 | 1.33 | **0.47** | 0.79 | 0.84 | 1.60 | 1.20 | **0.40** |
| Qwen3-8B | 0.78 | 0.60 | 0.95 | 0.58 | 2.18 | 1.90 | 0.28 | 0.96 | 0.91 | 2.00 | 1.71 | 0.29 |
| DeepSeek-R1 | 0.855 | 0.54 | 0.93 | 0.71 | 2.33 | 2.05 | 0.28 | 0.96 | 0.95 | 2.09 | 1.88 | 0.21 |
| GPT-4.1 | 0.895 | 0.44 | 0.90 | 0.60 | 2.30 | 2.13 | 0.17 | 0.90 | 0.62 | 2.13 | 1.94 | 0.19 |
| Gemini-2.5-pro | 0.925 | 0.42 | 0.86 | 0.57 | 2.19 | 1.90 | 0.29 | 0.89 | 0.91 | 2.15 | 1.61 | 0.54 |
| Mistral-medium-3 | 0.81 | 0.57 | 0.81 | 0.56 | 2.01 | 2.07 | -0.06 | 0.80 | 0.81 | 1.91 | 1.64 | 0.27 |
| Claude-sonnet-4 | 0.965 | 0.51 | 0.95 | 0.69 | 2.30 | 2.06 | 0.24 | 0.89 | 0.91 | 2.01 | 1.77 | 0.24 |
| Llama-3.3-70B-Instruct | 0.795 | 0.51 | 0.83 | 0.43 | 2.07 | 1.97 | 0.10 | 0.86 | 0.90 | 2.06 | 2.12 | -0.06 |

data, the higher threshold consistently yields better outcomes, indicating that the metric $C$ indeed selects contrastive pairs with clearer behavioral separation, making preference learning easier. Finally, comparing the 30k and 9k filtered sets shows that although high-quality samples help, too little data limits the model's ability to acquire abstract patterns such as elicitation adaptivity, whereas the 30k $C \geq 1$ set learns these behaviors effectively. This suggests that Once quality is ensured, sufficient data coverage remains essential for acquiring higher-level instructional strategies.

## J  HUMAN BASELINE ON PROPOSED GUIDEEVAL BENCHMARK

To establish a human baseline for comparison, we recruited an experienced frontline teacher with substantial real-world instructional and tutoring experience. The teacher was asked to provide responses on a subset of the evaluation data consisting of 400 instances—100 samples randomly selected from each of the four learner states. These teacher-produced responses were then evaluated using our framework, and the resulting human scores were compared against the performance of other models on the same 400-sample subset, as shown in Table 17.

With the human baseline incorporated, the behavioral differences between human teachers and LLMs become more pronounced. Human teachers excel at handling erroneous learner responses, providing explicit correction and fine-grained reconstruction—a natural product of professional teaching expertise—which leads to clear advantages in P-R and O-R. In contrast, current LLMs still struggle to deliver systematic error-aware remediation. At the same time, although human teachers tend to adopt a more conservative and steady instructional style—rarely pushing aggressively or posing frequent high-level questions—they nonetheless exhibit meaningful differences in question depth across the *accurate/erroneous* and *comprehension/confusion* conditions, adjusting their instructional challenge according to learner states. LLMs, despite often producing questions of generally higher nominal depth, show limited ability to modulate question difficulty based on the learner's cognitive condition, revealing a lack of genuine state-sensitive pedagogical adaptation.

## K  BROADER IMPACT

This work contributes a systematic benchmark for evaluating the guidance-oriented behaviors of LLMs in educational contexts. It underscores the necessity of designing models that can respond appropriately to diverse learners' cognitive states, promoting fairness and inclusivity in AI-driven education. As LLMs become more prevalent in classrooms, tutoring platforms, and remote learning environments, their capacity for context-sensitive, pedagogically sound guidance will directly affect educational quality, equity, and accessibility. Furthermore, this research lays the foundation for integrating longitudinal student interaction data, enabling personalized, trajectory-aware instruction that respects individual learning paths while safeguarding against potential biases.

## L  USE OF LLMs

In this study, large language models (LLMs) were employed as auxiliary tools to assist in the rewriting and refinement of selected text passages during the manuscript preparation process. All outputs generated by the models were carefully reviewed, edited, and filtered by the authors to ensure both accuracy and adherence to academic writing standards. It is important to note that the conceptual design, methodological development, data analysis, and interpretation of results were carried out independently by the authors without reliance on automated systems. The authors retain full responsibility for the originality, validity, and integrity of the work presented in this paper.

