# OpenReview forum: "Discerning Minds or Generic Tutors? Evaluating Instructional Guidance Capabilities in Socratic LLMs"
_ICLR.cc/2026/Conference — Submitted to ICLR 2026_

### Official Review · Reviewer_m5P2 · 2025-10-15

**Soundness:** 2
**Presentation:** 2
**Contribution:** 2
**Rating:** 4
**Confidence:** 4

**Summary:**

The paper introduces GuideEval, a benchmark to evaluate the tutoring capabilities of LLMs across three stages based on interactions with real students.
The authors use the benchmark to evaluate current commercial and open-source LLMs and find, for example, that they struggle to adapt to learner states, especially when critiquing wrong solutions.
The authors also use their data to finetune an LLM and show that finetuning improves these shortcomings.

**Strengths:**

- The research area is severely lacking resources that use real students (mainly due to ethical and regulatory problems / hurdles), so releasing a dataset of this scale with real students is quite impactful.

- The results show interesting observations for the future, for example, they point out the sycophancy of current LLMs which can be bad for learning.

**Weaknesses:**

- In general, the discussion of related work is a bit short and many findings of the paper have been discovered in prior works (though in different form). For example, Wang et al. 2023 already shows that LLMs tend to not actively engage with student error patterns and Daheim et al. 2024 show that identifying student mistakes is a challenging task even for sota LLMs. Another example is Dinucu-Jianu et al. 2025 who show that there exists a trade-off between pedagogy and giving hints (here the authors also point out that only focusing on not telling solutions leads to poor teaching). Scarlatos et al. 2025 also use DPO but this is not discussed, too.
I don't think that this takes away from the papers findings in general but proper discussion would improve it.
The authors also do not discuss knowledge tracing but the goal of this field is precisely what the paper sets out: to adapt to current learner states. I think a discussion of such concepts could also be helpful.

- I am wondering about more details about the dataset, many seem missing, as the discussion seems limited to Sec. 2.3.
For example, no annotator agreement is reported and beyond the number of turns and dialogues there are no further statistics. I think even reporting simple statistics would be helpful. The domain is also not specified beyond saying that it consists of middle school science questions. It is also mentioned that humans verified and even edited utterances (to create negatives) but the exact process is not detailed.

- The agreement between LLM and humans is fairly high but only checks binary preference over a combination of metrics and not agreement for a specific metric which limits expressiveness for how reliable the individual metrics are.

**Questions:**

- How many examples were used for creating Tab. 3?

- Will you release the dataset publicly? Based on the conclusion I would assume so but I did not find it written anywhere.

## References

Wang et al., Bridging the Novice-Expert Gap via Models of Decision-Making: A Case Study on Remediating Math Mistakes, NAACL 2024

Daheim et al., Stepwise Verification and Remediation of Student Reasoning Errors with Large Language Model Tutors, EMNLP 2024

Dinucu-Jianu et al., From Problem-Solving to Teaching Problem-Solving: Aligning LLMs with Pedagogy using Reinforcement Learning., arXiv 2025

Scarlatos et al., Training llmbased tutors to improve student learning outcomes in dialogues, AIED 2025

---

> ### Author Response · Authors · 2025-11-24
> **Response to reviewer m5P2 (1/2)**
>
> Thank you for your constructive comments.
>
> Here we address the points you mentioned in the weaknesses part.
> ---
> ---
>
> **1.** We appreciate the reviewer’s suggestion to strengthen the discussion of prior work.**In the revised manuscript, we add a subsection titled "LLM-based Knowledge Tracing"** which use LLMs to infer learner mastery, predict next-step correctness, or model state-aware tutoring. While they highlight the growing interest in state-sensitive instructional behaviors, our work is complementary: providing a structured, behavior-grounded evaluation framework targeting adaptivity, an aspect that remains under-evaluated.
>
> References:
>
> [1] Wang et al., Bridging the Novice-Expert Gap via Models of Decision-Making: A Case Study on Remediating Math Mistakes, NAACL 2024
>
> [2] Daheim et al., Stepwise Verification and Remediation of Student Reasoning Errors with Large Language Model Tutors, EMNLP 2024
>
> [3] Dinucu-Jianu et al., From Problem-Solving to Teaching Problem-Solving: Aligning LLMs with Pedagogy using Reinforcement Learning., arXiv 2025
>
> [4] Scarlatos et al., Training llmbased tutors to improve student learning outcomes in dialogues, AIED 2025
>
>
> ---
>
> **2.** We thank the reviewer for highlighting the need for more detals.
>
> (1)  **We added additional dataset statistics in Appendix D.2 in the revised version.** Specifically, the dataset contains 26,723 user utterances with 2,499,059 tokens. Figure 7 in the appendix illustrates the histogram of user-turn counts. At the token level, edited tokens constitute only 1.36% of the entire corpus, and at the turn level, edited turns account for 8.25% of all dialogue turns. These proportions indicate that state editing remains modest in scale, thereby preserving the overall linguistic characteristics of the original corpus.
>
> (2) We also clarify that the domain of the dialogues is mathematics. In the initial design stage, we consulted experienced in-service teachers, whose insights supported our current focus: (i) mathematics is the discipline that demands the most tutoring support, and correspondingly, most existing K–12 Socratic tutoring systems are math-centered; and (ii) mathematics, particularly at the middle-school level, offers a step-wise reasoning structure that makes learners’ cognitive states more observable, making it an appropriate starting point for evaluating adaptive guidance.
>
> (3) The state editing process is **LLM-based state rewriting followed by human check for contextual coherence and proper state alignment**. The LLM-based rewriting is as follows.
>
> + **Correct -> Incorrect:** We provided the LLM with the correct answer and instructed it to generate a plausible incorrect variant, preserving the student’s original reasoning and style.
> + **Comprehension <-> Confusion:** The LLM first classified the student’s original utterance, and then replaced it with authentic short expressions of “understanding” or “confusion”. These expressions were sampled from authentic student dialogues to maintain realism.
>
> **We added the state editing process in Appendix D.1 and the dataset statistics in Appendix D.2 in the revised version.**

---

> ### Author Response · Authors · 2025-11-24
> **Response to reviewer m5P2 (2/2)**
>
> **3.** We appreciate the reviewer’s concern regarding the granularity of agreement analysis. We would like to clarify that our evaluation already includes metric-level human–model consistency. Specifically, Table 3 reports agreement for each of the six instructional behavior dimensions independently, rather than aggregating them into a single binary preference.
>
> The consistency evaluation set was constructed from outputs of six LLMs (DeepSeek-V3, Qwen3-32B, Qwen3-8B, DeepSeek-R1-0528, o4-mini, and Spark X1) under four learner-state conditions. We randomly sampled 100 positive (score = 1) and 100 negative (score = 0) LLM-rated instances for each state in _Perception_ (accurate vs. erroneous) and _Orchestration_ (all four learner states). For _Elicitation_, we randomly sampled 100 instances for each of the three levels (1, 2, 3). This process yielded a total of 1,500 samples for human annotation. Each sample was then independently labeled by three human raters, with majority voting used to derive the final human label.
>
> To further assess label reliability, we additionally report Cohen’s $\kappa$ between every pair of raters for each evaluation dimension in below table. The agreement pattern exhibits a clear hierarchy—Perception > Orchestration > Elicitation—reflecting that more abstract instructional dimensions are inherently harder to operationalize and thus yield lower inter-rater consistency. Overall, the majority-voted human judgments demonstrate strong and stable alignment with our evaluation framework across all instructional dimensions.
>
> **We have updated all methodological details in Appendix C in the revised version.**
>
> | **Comparison** | **P-Affirm (n=200)** | **P-Redirect (n=200)** | **O-Advance (n=400)** | **O-Reconfigure (n=400)** | **Elicitation (n=300)** | **Overall Kappa** |
> | --- | --- | --- | --- | --- | --- | --- |
> | human1 vs human2 | 0.9200 | 0.9700 | 0.5550 | 0.7100 | 0.4700 | 0.7169 |
> | human1 vs human3 | 0.8900 | 0.9500 | 0.5850 | 0.5450 | 0.4850 | 0.6884 |
> | human2 vs human3 | 0.9100 | 0.9399 | 0.5627 | 0.6643 | 0.5911 | 0.7437 |
> | **Human majority vs Model** | **0.9200** | **0.9199** | **0.7378** | **0.7454** | **0.6631** | **0.8012** |
>
> ---
>
> Then here are the answers to the Questions part.
> ---
> ---
>
> **1.** We use 1500 samples to create Table 3. Please refer to our response to Weakness 3 for full information.
>
> ---
>
> **2.** Yes, we plan to release the dataset publicly. For transparency and reproducibility, the complete dataset has provided in the supplementary materials.

---

### Official Review · Reviewer_2FLd · 2025-10-31

**Soundness:** 3
**Presentation:** 4
**Contribution:** 3
**Rating:** 6
**Confidence:** 3

**Summary:**

This paper conducted an extensive evaluation of the capabilities of LLMs in guiding learners and dynamically adapting its responses to the learners' states. To achieve this, they also collected a benchmark dataset of real multi-turn dialogues of learners from a Socratic tutoring platform.

**Strengths:**

- The paper conducted an extensive analysis of various LLMs and provides several insights on their capability to recognize learner states, to guide / scaffold, and to elicit further follow-ups.
- The collected dataset GuideEval can help advance the field further.
- LLM-based scoring were validated with human annotations
- The failure analysis provides useful insights

**Weaknesses:**

- The authors evaluated the consistency between LLM based scoring and the Human annotators using the proportion of the same labels. I am not sure if this is the right way to go about it since simply showing the proportion of agreement can be misleading, especially if there is an imbalance in the label distribution. I believe there are more appropriate inter-rater agreement metrics that account for these.
- The failure analysis categorizes the types of failures but the authors did not seem to provide the frequencies of occurrence for each type of failure category.
- The authors only measured how the LLMs responded. For example, P-affirm and P-redirect scores are only based on how the LLM responded. But this does not tell us whether or not the LLM can recognize the learner states. It might be the case that the LLM does indeed recognize but just does not know how to respond properly.

**Questions:**

- How well can LLMs recognize the learner states (independent of how affirmative their responses are)?
- What is the distribution of failure types for each of the LLM models? Do they fail in similar ways?

**Details Of Ethics Concerns:**

The authors collected a dataset of learner dialogues. The responses were anonymized.
Not sure if this warrants an ethics review, but I'm filling this in since it seems to matches one of the categories (human subjects) for ethics review.

---

> ### Author Response · Authors · 2025-11-24
> **Response to reviewer 2FLd (1/2)**
>
> Thank you for your thoughtful comments and suggestions.
>
> Here we address the points you mentioned in the weaknesses part.
> ---
> ---
>
> **1.** We thank the reviewer for the insightful comment. We apologize for not clearly explaining the construction of our consistency evaluation set, which may have suggested that proportion agreement was affected by label imbalance. In fact, the set was deliberately balanced.
>
> The consistency evaluation set was constructed from outputs of six LLMs (DeepSeek-V3, Qwen3-32B, Qwen3-8B, DeepSeek-R1-0528, o4-mini, and Spark X1) under four learner-state conditions. We randomly sampled 100 positive (score = 1) and 100 negative (score = 0) LLM-rated instances for each state in DeepSeek-V3, Qwen3-32B, Qwen3-8B, DeepSeek-R1-0528, o4-mini, and Spark X1 under four learner-state conditions. We randomly sampled 100 positive (score = 1) and 100 negative (score = 0) LLM-rated instances for each state in _Perception_ (accurate vs. erroneous) and _Orchestration_ (all four learner states). For _Elicitation_, we randomly sampled 100 instances for each of the three levels (1, 2, 3). This process yielded a total of 1,500 samples for human annotation. Each sample was then independently labeled by three human raters, with majority voting used to derive the final human label.
>
> To further assess label reliability, we additionally report Cohen’s $\kappa$ between every pair of raters for each evaluation dimension in below table. The agreement pattern exhibits a clear hierarchy—Perception > Orchestration > Elicitation—reflecting that more abstract instructional dimensions are inherently harder to operationalize and thus yield lower inter-rater consistency. Overall, the majority-voted human judgments demonstrate strong and stable alignment with our evaluation framework across all instructional dimensions.
>
> **We have updated all methodological details in Appendix C in the revised version.**
>
> | **Comparison** | **P-Affirm (n=200)** | **P-Redirect (n=200)** | **O-Advance (n=400)** | **O-Reconfigure (n=400)** | **Elicitation (n=300)** | **Overall Kappa** |
> | --- | --- | --- | --- | --- | --- | --- |
> | human1 vs human2 | 0.9200 | 0.9700 | 0.5550 | 0.7100 | 0.4700 | 0.7169 |
> | human1 vs human3 | 0.8900 | 0.9500 | 0.5850 | 0.5450 | 0.4850 | 0.6884 |
> | human2 vs human3 | 0.9100 | 0.9399 | 0.5627 | 0.6643 | 0.5911 | 0.7437 |
> | **Human majority vs Model** | **0.9200** | **0.9199** | **0.7378** | **0.7454** | **0.6631** | **0.8012** |
>
>
> ---
>
>
> **2.** Thanks for good suggestion. We now selected two representative models, Qwen3-8B and GPT-4.1, and examined their failure cases across all instructional dimensions. For both positive learner states (accurate / comprehension) and negative learner states (erroneous / confusion), we randomly sampled 50 failure cases per state for each dimension (all with rubric scores of 0). This yields 100 analyzed cases per dimension per model. We then manually categorized each case according to our failure taxonomy and calculated the frequency distribution of all failure types. **The results are shown in the table below and more details in Appendix A.**
>
> We observe that compared with GPT-4.1, smaller models such as Qwen3-8B tend to exhibit more Rigid-related failures (e.g., repeated checkpointing or revisiting earlier steps), and are also more likely to mistakenly affirm incorrect student answers. Despite these differences, both models display similar weaknesses under erroneous or confused learner states, particularly in their lack of targeted remediation and their failure to introduce new instructional input. We believe this addition offers a clearer and more empirically grounded understanding of the failure patterns.
>
> | **Dimension** | **Error Scenario** | **Subtype** | **GPT-4.1** | **Qwen3-8B** |
> | --- | --- | --- | --- | --- |
> | **Perception** | Positive state failure (n=50) | Rigid adherence to procedural form | 6 | 14 |
> | | | Unwarranted skepticism toward foundational knowledge | 14 | 19 |
> | | | Failure to track student response intent | 6 | 2 |
> | | | Other | 24 | 15 |
> | | Negative state failure (n=50) | Uncritical acceptance of student answers | 21 | 26 |
> | | | Contradictory positive feedback | 12 | 10 |
> | | | Other | 17 | 14 |
> | **Orchestration** | Positive state failure (n=50) | Instructional misalignment caused by prior perception failure | 7 | 4 |
> | | | Rigid checkpointing despite comprehension | 20 | 27 |
> | | | Other | 23 | 19 |
> | | Negative state failure (n=50) | Lack of targeted remediation following vague feedback | 15 | 16 |
> | | | No new instructional input after confusion | 21 | 24 |
> | | | Other | 14 | 10 |

---

> ### Author Response · Authors · 2025-11-24
> **Response to reviewer 2FLd (2/2)**
>
> **3.** We thank the reviewer for raising this point. Our evaluation is behavior-based because, in real tutoring dialogue, state recognition and instructional response are not independent processes. A tutor (human or model) must infer the learner’s state before producing any pedagogical action, and this inference is necessarily reflected in the response itself. Thus, examining the response is an ecologically valid way to evaluate recognition in interactive settings.
>
> In our framework, the Perception dimension already separates recognition from strategy quality:
>
> + A score of 0 indicates an explicit misjudgment of the learner’s state.
> + Non-zero scores (0.5 or 1) indicate correct or partially correct recognition, after which Orchestration and Elicitation measure the appropriateness of the chosen strategy.
>
> To illustrate this point, we conducted an additional **explicit state-recognition experiment**. Using DeepSeek-V3, we fed the full dialogue context from our evaluation samples and asked the model to perform a four-way classification task: determining whether the student’s final turn belonged to _accurate_, _erroneous_, _comprehension_, or _confusion_ (mapped to labels 1–4). The experiment shows that the model’s explicit recognition accuracy on the accurate vs. erroneous classification task (0.8808) closely matches its non-zero Perception score ratio (0.8866) under our evaluation framework. Furthermore, the model performs strongly in distinguishing _comprehension_ vs. _confusion_ (0.9683), indicating stable performance in detecting explicit signals of understanding or uncertainty.
>
> ---
>
> Then here are the answers to the Questions part.
> ---
> ---
> **1.** Please refer to our response to Weakness 3 for full information.
>
> **2.**  Please refer to our response to Weakness 2 for full information.

---

### Official Review · Reviewer_y1Aq · 2025-10-31

**Soundness:** 3
**Presentation:** 3
**Contribution:** 3
**Rating:** 6
**Confidence:** 3

**Summary:**

This paper introduces GuideEval, a benchmark for evaluating instructional guidance capabilities of large language models (LLMs) when serving as Socratic tutors. The authors argue that existing evaluations focus primarily on question generation while overlooking adaptive guidance—the ability to dynamically adjust teaching strategies based on learners' cognitive states. The paper proposes a three-phase behavioral framework: (1) Perception - inferring learner states (accurate/erroneous/comprehension/confusion); (2) Orchestration - adapting instructional strategies through scaffolding; and (3) Elicitation - stimulating deeper thinking through strategic questioning.

The authors construct a dataset of 5,177 test samples from authentic tutoring dialogues with contrastive student state pairs, enabling controlled evaluation of model adaptivity. They evaluate 14 LLMs across 6 metrics derived from their framework, finding that models struggle with error detection, adapting to implicit cognitive cues, and maintaining consistent guidance strategies. The paper includes detailed failure pattern analysis and demonstrates that behavior-guided finetuning with Chain-of-Thought distillation substantially improves guidance performance.

**Strengths:**

1. The paper focuses on critical gap in LLM tutoring evaluation by focusing on adaptive guidance rather than static content quality.

2. The three-phase model is well-motivated by educational psychology literature and operationalized into measurable metrics.

3. The exp covers 14 diverse models, revealing consistent failure patterns across architectures.

4.The paper contains a detailed failure case analysis, providing an intuitive understanding beyond quantitative metrics.

5. I really like the comparative analysis of different training strategies part. The finding that outcome-only SFT degrades performance while process supervision (CoT Distillation) and pairwise preference optimization (DPO) provide substantial gains is a critical, actionable insight for the community.

**Weaknesses:**

1. it comes with limited scope: dataset topic - middle school science problems in Chinese. It would be more curated if you expand it to other difficulty levels and languages.

2. The cognitive modeling with 4 states (Accurate, Erroneous, Comprehension, Confusion) may be too simplified to capture nuanced learning states. As authors acknowledge, it doesn't capture individual learner profiles, misconception history, or engagement patterns

**Questions:**

1. Have you tested or do you plan to test this framework on other domains (e.g., humanities, programming) or age groups? What challenges do you anticipate?

2. Can you provide comparison with human tutor performance on the same benchmark? This would contextualize LLM performance.

3. The "Designed Prompt" used to generate the training data for the finetuning experiments is very explicit and rule-based. How can we be sure that the model learned a generalizable instructional guidance capability rather than just learning to mimic the explicit rules baked into the generation prompt? Have you tested or are you planning to test the finetuned model on out-of-domain tasks to see if the "skill" transfers?

4. The filtering mechanism for training data (equations on p.18) is complex. How sensitive are results to these hyperparameters (β, threshold)?

---

> ### Author Response · Authors · 2025-11-24
> **Response to reviewer y1Aq (1/2)**
>
> Thank you for your valuable feedback.
>
> Here we address the points you mentioned in the weaknesses part.
> ---
> ---
>
> **1.** We appreciate the reviewer’s suggestion to expand GuideEval to additional difficulty levels and languages. The core objective of GuideEval, evaluating behavior-level instructional guidance, is inherently language/domain-agnostic, as our behavioral framework (perception–orchestration–elicitation) does not depend on linguistic properties or domain-specific content. In the initial design stage, we consulted experienced in-service teachers, whose insights supported our current focus: (1) mathematics is the discipline that demands the most tutoring support, and correspondingly, most existing K–12 Socratic tutoring systems are math-centered; and (2) mathematics, particularly at the middle-school level, offers a step-wise reasoning structure that makes learners’ cognitive states more observable, making it an appropriate starting point for evaluating adaptive guidance. Our decision to build the benchmark in Chinese reflects practical constraints in acquiring high-quality, authentic learner-state annotations. We fully agree that expanding to other languages and academic levels would be beneficial, and we plan to extend GuideEval accordingly in future work.
>
> ---
>
> **2.** We appreciate the reviewer’s point. The four states (Accurate, Erroneous, Comprehension, and Confusion) capture the core contingencies that expert tutors routinely respond to and are sufficient to evaluate the specific capability we target: whether an LLM can perceive essential pedagogical cues and adapt its guidance accordingly. While richer dimensions such as learner profiles or misconception histories are valuable in broader educational research, incorporating them into a benchmark would reduce annotation reliability and introduce confounds that hinder controlled evaluation. We agree that future extensions can incorporate finer-grained learner characteristics. For the present benchmark, the four-state model provides an effective balance between fidelity, interpretability, and feasibility, and this simplified structure already reveals substantial deficiencies in current LLMs’ adaptive scaffolding, highlighting an important area for model developers to address.
>
> ---
>
> Then here are the answers to the Questions part.
> ---
>
> ---
> 1. Thanks for suggestion. While GuideEval is currently grounded in authentic K–12 mathematics dialogues (a deliberate choice informed by consulting experienced in-service teachers, who highlighted that mathematics requires extensive tutoring support and offers a step-wise reasoning structure that makes learners’ cognitive states more observable), we believe the framework is inherently generalizable to other domains, including humanities and programming, and to different learner age groups. We think the primary challenge in extending to new domains lies in designing domain-specific scaffolding strategies, which may require adapting behavioral prompts to reflect the particular reasoning or expressive patterns of each subject and for different age groups. A second challenge is obtaining high-quality, authentic tutoring dialogues that accurately capture real pedagogical interactions. We plan to investigate these extensions in future work by collecting diverse domain-specific dialogues and validating the framework’s adaptability across varied instructional contexts.
>
> ---
>
> 2. We thank the reviewer for highlighting the value of human–tutor comparison. To address this, we are currently recruiting in-service teachers to respond to a subset of held-out GuideEval contexts under the same experimental conditions as the LLMs. Their responses will be evaluated using the same three-phase behavioral rubric, allowing a direct comparison across the six instructional dimensions.
>
> This evaluation is still in progress. **We plan to complete it prior to the rebuttal deadline and will incorporate the results or a detailed summary into the revised manuscript.**

---

> ### Author Response · Authors · 2025-11-24
> **Response to reviewer y1Aq (2/2)**
>
> **3.** This is a good question. We would like to highlight two aspects regarding generalization:
>
> **(1)** DPO and CoT distillation target high-level instructional behaviors rather than memorizing specific prompt patterns. CoT distillation, in particular, trains the model to reproduce the teacher’s reasoning and strategic decisions, which are context-dependent and not rule-based.
>
> **(2)** In contrast, Response-only finetuning, which exposes the model mainly to output templates, leads to performance degradation, indicating that imitating structured formats or rule-like outputs does not enhance instructional ability. CoT distillation, however, improves performance across behavioral dimensions, demonstrating that the model internalizes teaching strategies rather than memorizing prompts.
>
> These results indicate that the finetuned model has learned generalizable instructional guidance beyond the examples in the designed prompt.
>
> ---
>
>
> **4.** We thank the reviewer for the comment. First, the score differences `A_delta` and `E_delta` come from **very small discrete sets**:
> + Perception scores ∈ `{0, 0.5, 1}`; Orchestration scores ∈ `{0, 1}`.
> + ESA differences `E_delta` lie in `{-3, -2, -1, 0, 1, 2, 3}`.
>
> Thus, the combined score
> `C = A_delta * (1 + beta * max(E_delta, 0))`
> can only take a limited set of discrete values, making the filtering naturally can only take a limited set of discrete values, making the filtering naturally **insensitive** to small changes in `beta`. Thus, varying beta within a reasonable range (e.g., 0.1–0.3) does not affect which samples pass the threshold, as the auxiliary term is always secondary to the main A_delta term. We also show the empirical distributions of C **in Appendix E.2 (Figure 8) ** , showing that choosing the threshold C≥1 effectively preserves sufficient meaningful instructional-behavior contrasts while filtering out noise. If the threshold was set substantially lower, more noisy samples would be introduced; if set substantially higher, the retained data volume would shrink—both scenarios would weaken finetuning performance.  Overall, the filtering procedure is discrete, stable, and semantically motivated. **We will report ablation results under different values of C before the rebuttal deadline.**

---

> ### Author Response · Authors · 2025-11-29
> **Response to reviewer y1Aq**
>
> Here we provide further response including analysis and additional experimental results to Questions 2 and Questions 4.
> ----
>
> 1. **Further response to Questions 2.**
>
> According to your suggestion, we now have added an evaluation of human baseline in **Appendix J: The Performance of Human Teachers in the Evaluation**. We recruited an experienced frontline teacher (due to time limit) and collected responses on a 400-sample subset (100 instances per learner state), and evaluated these responses using the same rubric as the LLMs.
>
> **The results in Table 16 in the revised manuscript** reveal clear behavioral differences: human teachers excel in handling erroneous learner inputs, showing strong corrective feedback and reconstruction (high P-R and O-R), whereas current LLMs still struggle to provide systematic error-aware remediation. Meanwhile, although human teachers adopt a more conservative teaching style, they nevertheless adjust question depth meaningfully across learner states, while LLMs—despite often producing higher-level questions—show limited ability to modulate question difficulty based on student state.
>
> 2. **Further response to Questions 4.**
>
> **We conducted a more systematic ablation study on the filtering threshold (C), and the results have been added to Appendix I.** As shown in the table, we compared four datasets for DPO fine-tuning Qwen3-8B under identical settings: (C ≥ 0), (C ≥ 1), (C ≥ 2), and a 9k subset randomly sampled from the (C ≥ 1) pool with matched pair-type proportions and the same size as the (C ≥ 2) set.
>
> The results indicate that lowering the threshold from (C ≥ 1) to (C ≥ 0) leads to clear degradation, suggesting that weak filtering introduces conflicting preference signals into DPO. With equal 9k samples, the (C ≥ 2) dataset consistently outperforms the sampled 9k subset, confirming that (C) effectively selects more contrastive and reliable pairs. Moreover, both 9k datasets struggle to learn abstract behaviors such as ESA, whereas the full 30k (C ≥1) dataset shows substantial improvements. This demonstrates that once quality is ensured, sufficient data coverage remains essential for learning higher-level instructional strategies.
>
> For  Accurate / Erroneous
>
> | **Model** | **P-A ↑** | **P-R ↑** | **O-A ↑** | **O-R ↑** | **E-S ↑** | **E-H** | **ESA ↑** |
> | --- | --- | --- | --- | --- | --- | --- | --- |
> | **Qwen3-8B (baseline)** | 0.7613 | 0.5919 | 0.9176 | 0.6681 | 2.0605 | 1.9213 | 0.1049 |
> | **C ≥ 0 (49k)** | 0.9055 | 0.5093 | 0.7269 | 0.3169 | 1.7017 | 1.5458 | 0.1296 |
> | **C ≥ 1 (30k)** | 0.8357 | 0.8326 | 0.8672 | 0.7390 | 2.0361 | 1.7831 | 0.2667 |
> | **C ≥ 1 (9k)** | 0.9185 | 0.4564 | 0.7471 | 0.2907 | 1.7345 | 1.7356 | -0.0358 |
> | **C ≥ 2 (9k)** | 0.9055 | 0.5974 | 0.9294 | 0.4911 | 2.1664 | 2.1041 | 0.0605 |
>
>
> For   Comprehension / Confusion
>
> | **Model** | **O-A ↑** | **O-R ↑** | **E-S ↑** | **E-H** | **ESA↑** |
> | --- | --- | --- | --- | --- | --- |
> | **Qwen3-8B (baseline)** | 0.9644 | 0.8959 | 2.0527 | 1.7254 | 0.3281 |
> | **C ≥ 0 (49k)** | 0.6793 | 0.6579 | 1.5103 | 1.0870 | 0.4230 |
> | **C ≥ 1 (30k)** | 0.9565 | 0.9743 | 2.1775 | 1.7263 | 0.4512 |
> | **C ≥ 1 (9k)** | 0.6857 | 0.5303 | 1.6151 | 1.2694 | 0.3457 |
> | **C ≥ 2 (9k)** | 0.9415 | 0.9408 | 2.1419 | 1.9636 | 0.1719 |

---

### Official Review · Reviewer_QMx2 · 2025-11-01

**Soundness:** 2
**Presentation:** 3
**Contribution:** 2
**Rating:** 4
**Confidence:** 3

**Summary:**

This paper investigates whether LLM tutors can deliver adaptive instructional guidance in Socratic dialogues rather than relying on generic questioning. The authors conceptualize guidance as a three-phase behavior: Perception (inferring the learner’s state), Orchestration (selecting an appropriate next-step strategy), and Elicitation (formulating prompts suited to the learner’s state). Building on this framework, they introduce GuideEval, a benchmark composed of real multi-turn tutoring dialogues with contrastive student states, and define phase-aligned evaluation metrics to assess model performance across these dimensions.

**Strengths:**

1. Clear behavioral decomposition with actionable metrics. The three-phase split translates “be a better tutor” into concrete, checkable behaviors, offering conceptual clarity and operational guidance that enable reproducible, phase-wise diagnosis across different models.

2. Useful failure taxonomy grounded in qualitative evidence. The paper goes beyond reporting average behaviors and highlights failure modes supported by dialogue snippets, providing interpretability and practical insight into model behavior.

**Weaknesses:**

1. Human–LLM agreement is reported without sample size or reliability statistics. In Table 3, the claim that “LLMs can serve as reliable and scalable evaluators of instructional behaviors” rests on high agreement ratios and minimal score deviations, but the paper omits sample size per metric or level, sampling protocol, number of human raters, and inter-rater reliability. Without these, chance agreement and selection bias cannot be ruled out, especially with coarse labels (binary or 3-point) that inflate raw agreement.

2. Prompt–rubric inconsistency. Generation prompts forbid giving final answers (“Do not directly provide the final answer or full solution process” in Original/Rule templates), yet the O-Advance evaluation rubric awards full credit when “the model provides the final answer.” This contradiction allows models to achieve high orchestration scores while violating generation rules. Please align the prompt and rubric definitions.

3. All models are decoded at temperature = 0.1. Please justify this setting and provide an ablation across temperatures to ensure conclusions are not artifacts of low-variance decoding.

4. Human annotators reportedly revised model outputs to create “state-edited counterparts” (e.g., answer-correctness flips, comprehension/confusion flips). Please specify the exact editing operations and provide per-operation statistics, including counts, edit distance, token-level change distribution, and human vs. synthetic proportions.

**Questions:**

1. What are the details of sample size per metric or level, sampling protocol, number of human raters, and inter-rater reliability statistics?

2. Generation prompts forbid giving final answers, while the O-Advance rubric rewards them. How are these conflicting criteria reconciled to ensure consistent evaluation?

3. How many items per metric were used for the LLM–human consistency analysis, and how were they sampled from the full dataset?

4. Could you replicate headline results using at least one alternative judge and report inter-judge agreement and sensitivity to confirm robustness?

5. Since training and evaluation data are drawn from the same source pool, do problem contexts (e.g., identical or near-duplicate problem IDs, stems, passages, or scaffolds) repeat across splits? If so, what is the overlap rate, and how does it affect evaluation outcomes?

---

> ### Author Response · Authors · 2025-11-24
> **Response to reviewer QMx2 (1/2)**
>
> Thank you for taking the time to review our paper.
>
> Here we address the points you mentioned in the Weaknesses part.
> ---
> ---
>
> **1.** We thank the reviewer for raising this important point and provide the missing procedural details below. The consistency evaluation set was constructed from outputs of six LLMs (DeepSeek-V3, Qwen3-32B, Qwen3-8B, DeepSeek-R1-0528, o4-mini, and Spark X1) under four learner-state conditions. Specifically, we randomly sampled 100 positive (score = 1) and 100 negative (score = 0) LLM-rated instances for each state in _Perception_ (accurate vs. erroneous) and _Orchestration_ (all four learner states). For _Elicitation_, we randomly sampled 100 instances for each of the three levels (1, 2, 3). This process yielded a total of 1,500 samples for human annotation. Each sample was then independently labeled by three human raters, with majority voting used to derive the final human label.
>
> To further assess label reliability, we additionally report Cohen’s $\kappa$ between every pair of raters for each evaluation dimension in below table. The agreement pattern exhibits a clear hierarchy—Perception > Orchestration > Elicitation—reflecting that more abstract instructional dimensions are inherently harder to operationalize and thus yield lower inter-rater consistency. Overall, the majority-voted human judgments demonstrate strong and stable alignment with our evaluation framework across all instructional dimensions.
>
> **We have updated all methodological details in Appendix C in the revised version.**
>
> | **Comparison** | **P-Affirm (n=200)** | **P-Redirect (n=200)** | **O-Advance (n=400)** | **O-Reconfigure (n=400)** | **Elicitation (n=300)** | **Overall Kappa** |
> | --- | --- | --- | --- | --- | --- | --- |
> | human1 vs human2 | 0.9200 | 0.9700 | 0.5550 | 0.7100 | 0.4700 | 0.7169 |
> | human1 vs human3 | 0.8900 | 0.9500 | 0.5850 | 0.5450 | 0.4850 | 0.6884 |
> | human2 vs human3 | 0.9100 | 0.9399 | 0.5627 | 0.6643 | 0.5911 | 0.7437 |
> | **Human majority vs Model** | **0.9200** | **0.9199** | **0.7378** | **0.7454** | **0.6631** | **0.8012** |
>
>
> ---
>
> **2.** The perceived contradiction does not arise in our pipeline. Generation prompts are used solely to produce tutoring trajectories, while the evaluation rubric is applied afterward; models do not have access to the rubric during generation and thus cannot “violate” prompt rules. Regarding credit for producing the final answer, such credit is awarded only when the answer emerges as a legitimate closure move at the end of a multi-turn reasoning process, consistent with established tutoring practice [1], rather than as premature answer-giving. This distinction ensures a fair and principled assessment of orchestration quality.
>
> **Reference**
>
> [1] Graesser, A. C., Person, N. K., & Magliano, J. (1995). _Collaborative dialog patterns in naturalistic one-on-one tutoring_. Applied Cognitive Psychology, 9(6), 495–522.
>
> ---
>
> **3.** We appreciate the reviewer’s concern regarding the decoding temperature. We set the temperature to 0.1 to ensure low-variance, deterministic outputs, reducing sampling variance and yielding responses that reflect the model’s “average” pedagogical tendencies. In educational contexts, stability and correctness are prioritized over creative diversity. At the same time, setting temperature slightly above zero avoids degenerate repetitive patterns, preserving natural linguistic variation characteristic of human tutors. To validate robustness, we performed an ablation study with Qwen3-8B at temperatures 0, 0.1, 0.5, and 1.0 and the results are shown below. While minor fluctuations in scores were observed, the overall trends in guidance effectiveness remain unchanged.
>
> **We included these results in the Appendix G.3 of revised manuscript to clarify the effect of decoding temperature.**
>
> For Acc/Err state
> | Temperature | P-A ↑ | P-R ↑ | O-A ↑ | O-R ↑ | E-S ↑ | E-H | ESA ↑ |
> | --- | --- | --- | --- | --- | --- | --- | --- |
> | **0** | 0.7649 | 0.5513 | 0.9092 | 0.6226 | 2.0706 | 1.9432 | 0.1397 |
> | **0.1** | 0.7613 | 0.5919 | 0.9176 | 0.6681 | 2.0605 | 1.9213 | 0.1049 |
> | **0.5** | 0.7550 | 0.5720 | 0.9143 | 0.6602 | 2.0790 | 1.9644 | 0.1099 |
> | **1.0** | 0.7227 | 0.5538 | 0.8933 | 0.6415 | 2.0630 | 2.0102 | 0.0481 |
>
> For Comp/Conf state
> | Temperature | O-A ↑ | O-R ↑ | E-S ↑ | E-H | ESA ↑ |
> | --- | --- | --- | --- | --- | --- |
> | **0** | 0.9508 | 0.8916 | 2.1320 | 1.7703 | 0.3612 |
> | **0.1** | 0.9644 | 0.8959 | 2.0527 | 1.7254 | 0.3281 |
> | **0.5** | 0.9487 | 0.8774 | 2.0870 | 1.8019 | 0.2851 |
> | **1.0** | 0.9408 | 0.8603 | 2.1426 | 1.8617 | 0.2808 |

---

> ### Author Response · Authors · 2025-11-24
> **Response to reviewer QMx2 (2/2)**
>
> **4.** We thank the reviewer for the question. The state editing process is **LLM-based state rewriting followed by human check for contextual coherence and proper state alignment**. The LLM-based rewriting is as follows.
> + **Correct -> Incorrect:** We provided the LLM with the correct answer and instructed it to generate a plausible incorrect variant, preserving the student’s original reasoning and style.
> + **Comprehension <-> Confusion:** The LLM first classified the student’s original utterance, and then replaced it with authentic short expressions of “understanding” or “confusion”. These expressions were sampled from authentic student dialogues to maintain realism.
>
> At the token level, edited tokens constitute only 1.36% of the entire corpus, and at the turn level, edited turns account for 8.25% of all dialogue turns. These proportions indicate that state editing remains modest in scale, thereby preserving the overall linguistic characteristics of the original corpus.
>
> **We added the state editing process in Appendix D.1 and the dataset statistics in Appendix D.2 in the revised version.**
>
> ---
>
> Then here are the answers to the Questions part.
> ---
> ---
>
> **1.** We have updated all methodological details to Appendix C in the revised version. Please refer to our response to Weakness 1 for full information.
>
> ---
>
> **2.** The perceived contradiction does not arise in our pipeline. Please refer to our response to Weakness 2 for full information.
>
> ---
>
> **3.** We randomly sampled 100 positive (score = 1) and 100 negative (score = 0) LLM-rated instances for each state in _Perception_ (accurate vs. erroneous) and _Orchestration_ (all four learner states). For _Elicitation_, we randomly sampled 100 instances for each of the three levels (1, 2, 3). This process yielded a total of 1,500 samples for human annotation to validate LLM-human consistency. Please refer to our response to Weakness 1 for more detail.
>
> ---
>
> **4.** We conducted an additional robustness experiment using **Claude-3-Haiku** as an alternative judge. We selected six representative models with clearly behavioral profiles in the main results (Qwen3-8B, DeepSeek-R1, GPT-4.1, Gemini-2.5-Pro, Mistral-Medium-3, and SocraticLM). The rank changes between the original evaluation (GPT-4o-mini) and the alternative judge, together with the Spearman correlations for each metric, are shown in below table. The rank shifts are small, and most Spearman correlations exceed 0.88, indicating strong consistency.
>
> **We added this analyais in Appendix G.2 in the revised manuscript, along with the detailed performance of the six models evaluated using the alternative judge on each metric in Table 11 and Table 12**, which shows that absolute scores differ slightly but the relative trends remain stable (e.g., GPT-4.1’s stronger higher-order questioning, SocraticLM’s elevated explicit affirmation).
>
> For Acc/Err state
> | Model | P-A ↑ | P-R ↑ | O-A ↑ | O-R ↑ | E-S ↑ | E-H | ESA ↑ |
> | --- | --- | --- | --- | --- | --- | --- | --- |
> | Qwen3-8B        | 0  | 0  | -1 | 0 | 1  | 1  | 0 |
> | DeepSeek-R1     | 0  | 1  | 0  | 0 | 0  | 0  | 0 |
> | GPT-4.1         | 0  | -1 | 1  | 0 | 0  | 0  | 0 |
> | Gemini-2.5-pro  | -1 | 1  | 0  | 0 | -1 | 0  | 0 |
> | mistral-medium-3| 0  | -1 | 0  | 0 | 0  | -1 | 0 |
> | SocraticLM      | 1  | 0  | 0  | 0 | 0  | 0  | 0 |
> | **Spearman ρ**  | 0.9429 | 0.8857 | 0.9429 | 1.0000 | 0.9429 | 0.9429 | 1.0000 |
>
>
> For Comp/Conf state
> | Model | O-A ↑ | O-R ↑ | E-S ↑ | E-H | ESA ↑ |
> | --- | --- | --- | --- | --- | --- |
> | Qwen3-8B        | -1 | 0 | 1  | 0  | 1 |
> | DeepSeek-R1     | 1  | 0 | 1  | 1  | -1 |
> | GPT-4.1         | -1 | 0 | -2 | -1 | 2  |
> | Gemini-2.5-pro  | 1  | 0 | 0  | 0  | -1 |
> | mistral-medium-3| 0  | 0 | 0  | 0  | -1 |
> | SocraticLM      | 0  | 0 | 0  | 0  | 0  |
> | **Spearman ρ**  | 0.8857 | 1.0000 | 0.8986 | 0.9559 | 0.7714 |
>
>
> ---
>
>
> **5.** Thanks for good question. We confirm that the construction of GuideEval strictly prevents any overlap between training and evaluation data. Specifically, all data splits were executed at the _problem-stem level_: each unique problem ID, together with all associated instructional dialogues, learner–tutor interactions, and constructed cognitive-state annotations, was assigned entirely to either the training pool or the evaluation benchmark _prior_ to any further processing. This guarantees that no problem stem, scaffold, passage, or near-duplicate variant appears across multiple splits. Consequently, the evaluation outcomes are not affected by shared contexts or implicit memorization, and the benchmark faithfully measures models’ generalization to unseen instructional scenarios.

---

### Author Response · Authors · 2025-12-03
**Rebuttal Summary**

To the newly assigned Area Chair,

Since reviewers can no longer join discussions or update scores, we sincerely thank them for their initial evaluations, which greatly helped improve our manuscript. We are especially grateful to the newly assigned Area Chair for taking on this responsibility. To support evaluation, we have included a “Rebuttal Summary” below, outlining the main revisions and responses addressing the reviewers’ concerns.

This paper introduces GuideEval, a benchmark for evaluating LLMs’ instructional guidance as Socratic tutors, shifting the focus from generic question generation to state-adaptive teaching behaviors. We formalize guidance into a three-phase framework and build metrics using real multi-turn dialogues with contrastive learner states. Using GuideEval, we show that current LLMs struggle particularly with erroneous and confused learners. We further provided comprehensive faliure analysis and introduced behavior-guided finetuning that partially alleviate these limitations.


**Key strengths highlighted by reviewers include:**

+ **A well-motivated and behaviorally grounded formulation of instructional guidance**, where the proposed three-phase framework offers clear and verifiable metrics for evaluating tutor-like behaviors (QMx2, y1Aq).
+ **A dataset derived from real multi-turn student interactions**, which reviewers noted as valuable given the scarcity of such resources in educational settings (2FLd, m5P2).
+ **Detailed failure analysis**, including the failure taxonomy and dialogue-based case studies, which reviewers noted as helpful for understanding model errors beyond numerical scores (QMx2, y1Aq).
+ **Extensive evaluation across diverse LLMs**, providing broad and interesting observations about how commercial and open-source models behave under different learner states (y1Aq, 2FLd, m5P2).
+ **Analysis of training strategies**, including comparisons showing differences between outcome-only SFT, CoT-distillation SFT, and preference-based optimization (y1Aq).


**In response to reviewer concerns, we provided the following clarifications and additional results:**


+ **Human–LLM consistency experimental details (QMx2, 2FLd, m5P2):** We added missing information on sample sizes, sampling protocol, number of human raters, and inter-rater reliability in Appendix C (Table 8).
+ **Dataset & State editing details (QMx2, m5P2):** We expanded the descriptions of correctness flips, comprehension/confusion flips, edit statistics, and dataset-level information in Appendix D.1 and D.2.
+ **Ablations on decoding temperatures (QMx2):** We evaluated multiple decoding temperatures (results in Table 14) to ensure that the findings are not artifacts of temperature sampling.
+ **Ablations on the judge model (QMx2):**  We repeated the evaluation using a different LLM judge (Table 11–13) to verify the robustness of the scoring framework.
+ **Human tutor baseline (y1Aq):**  We collected real teacher responses, evaluated them using our framework, and compared them against LLM performance on the same subset (Table 17) to contextualize model behaviors.
+ **Ablations on the filtering mechanism for training data (y1Aq) :** We analyzed the sensitivity of training results to the filtering thresholds values, with results summarized in Table 16.
+ **Failure-pattern distribution (2FLd) :** We added frequency statistics for failure types (Table 5) and compared failure patterns across models.
+ **The learner states recognition accuracy (2FLd) :**  We added a dedicated experiment on explicit learner-state recognition accuracy, providing complementary evidence that the Perception dimension relates to how models recognize learner states.
+ **Additional clarifications**
    - **prompt-rubic inconsistency (QMx2) :** We clarified the alignment between generation prompts and evaluation rubrics.
    - **rational for dataset domain and difficulty level selection (y1Aq, m5P2) :** We explained the motivation behind selecting middle-school mathematics as the domain.
    - **the generalization ability of the finetuned model (y1Aq) :**  We discussed whether the finetuned model learns generalizable instructional strategies beyond prompt imitation.
    - **discussion with knowledge tracing related work (m5P2) :**  We expanded the related work section to connect our benchmark with knowledge-tracing literature and clarify their differences.



The reviewers’ concerns mainly center on experimental details—especially the robustness of the evaluation protocol, the transparency of dataset construction, and the validity of human–LLM consistency measurements. Across all raised points, we have provided additional experiments, expanded analyses, and thorough clarifications. These additions further strengthen the rigor, reliability, and completeness of our work.

---

### Meta-Review · Area_Chair_xFvh · 2026-01-04

**Summary:**

It is pointed out that Socratic teaching lacks the ability to guide students based on their cognitive states, and an evaluation benchmark called Guide Eval is proposed. A fine-tuned language model has been tested on multiple benchmarks.
Reviewers find the three-stage evaluation approach reasonable and agree that the experiments are extensive and thorough.
The main concerns raised by the reviewers are as follows:
1. There are issues with the rigor and missing details in the evaluation, including inconsistencies between human and LLM evaluators, and contradictions in prompts and scoring criteria.
2. Insufficient details regarding data and methodology, including inadequate disclosure of specifics and risks related to data construction and data splitting.
3. Concerns about the limited coverage of dataset domains and the model's generalization ability.

**Reviewer Concerns:**

The author has supplemented extensive experiments, effectively addressing concerns such as parameter sensitivity. However, regarding issues like insufficient coverage of dataset domains, concerns about model generalization ability, and worries about dataset evaluation methods and quality, the author engaged in discussion but found it difficult to take substantive actions to resolve these concerns.

**Reviewer Scores:**

If thorough discussion is conducted, the reviewers are inclined to maintain their current scores.

---

### Decision · Program_Chairs · 2026-01-26

Reject